# SINGLE-STEP DIFFUSION MODEL-BASED GENERATIVE MODEL INVERSION ATTACKS

## ABSTRACT

Generative model inversion attacks (MIAs) have garnered increasing attention for their ability to reconstruct synthetic samples that closely resemble private training data, exposing significant privacy risks in machine learning models. The success of generative MIAs is primarily attributed to image priors learned by generative adversarial networks (GANs) on public auxiliary data, which help constrain the optimization space during the inversion process. However, GAN-based generative MIAs still face limitations, particularly regarding the instability during model inversion optimization and the fidelity of reconstructed samples, indicating substantial room for improvement. In this paper, we address these challenges by exploring generative MIAs based on diffusion models, which offer superior generative performance compared to GANs. Specifically, we replace the GAN generator in existing generative MIAs with a single-step generator distilled from pretrained diffusion models, constraining the search space to the manifold of the generator during the inversion process. In addition, we leverage generative model inversion techniques to investigate privacy leakage issues in widely used large-scale multimodal models, particularly CLIP, highlighting the inherent privacy risks in these models. Our extensive experiments demonstrate that single-step diffusion models-based MIAs significantly outperform their GAN-based counterparts, achieving substantial improvements in traditional metrics and greatly enhancing the visual fidelity of reconstructed samples. This research uncovers privacy vulnerabilities in CLIP models and opens new research directions in generative MIAs. Our source code is available at this anonymous repository: https://anonymous.4open.science/r/DDMI-F967/.

## 1 INTRODUCTION

As machine learning (ML) models advance and become increasingly integrated into critical domains such as healthcare diagnostics (Richens et al., 2020), intelligent finance (Rundo et al., 2019), and biometric authentication (Jain & Nandakumar, 2012), concerns about data privacy have grown. These models, particularly deep neural networks (DNNs), rely on sensitive datasets for training, making them appealing targets for adversarial attacks. One emerging threat is model inversion attacks (MIAs) (Fredrikson et al., 2014), a category of privacy attack, in which adversaries reconstruct samples to expose sensitive information from the private training data (*e.g.,* personally identifiable images) by analyzing the model's outputs. This poses a significant risk to user privacy and security.

Earlier studies (Fredrikson et al., 2015) framed MIAs as an optimization problem in the raw input space, aiming to reconstruct private data by adjusting synthetic inputs to maximize the likelihood of a specific class. This method was based on the assumption that strong correlations were established between inputs and model outputs during the training process. While this approach proved effective for simpler ML models, it was insufficient for DNNs trained on high-dimensional data, such as facial recognition models. In such cases, optimizing in the input space becomes highly complex and often ill-posed, leading to the generation of unrealistic features that lack semantic relevance.

To address this challenge, Zhang et al. (2020) proposed a novel approach called generative model inversion attacks (GMI), which enhances the inversion process by learning a meaningful image prior from public auxiliary data using generative adversarial networks (GANs) (panel (a) of Fig. 2). This ensures that the synthetic data generated during the attack resides on a realistic image manifold

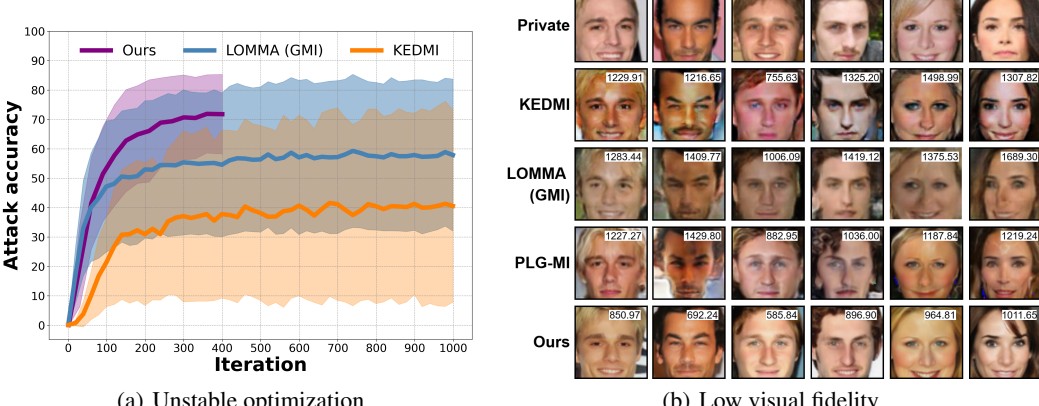

(a) Unstable optimization  (b) Low visual fidelity

Figure 1: **Limitations in GAN-based generative MIAs.** "Ours" refers to LOMMA (GMI) based on single-step diffusion models (Sec.3.3). (a) The attack accuracy improves in SOTA GAN-based MIAs as the number of optimization iterations increases. However, consistent fluctuations indicate instability in the optimization process. (b) Visualization of reconstructed images produced by SOTA GAN-based MIAs reveal low visual fidelity, leading to suboptimal inversion performance. The k-nearest neighbors distance (KNN Dist) for each image is shown in the top-right corner of the image. For detailed setups and additional results of the motivating experiments, refer to Appx. C.6.

(panel (c) of Fig. 2). This method has sparked significant advancements in GAN-based generative MIAs (Zhang et al., 2020; Chen et al., 2021; Wang et al., 2021a; Struppek et al., 2022; Yuan et al., 2023; Nguyen et al., 2023), allowing for more accurate reconstruction of samples that closely resemble the original private training data, thus driving substantial progress in this field.

While GAN-based MIAs outperform traditional methods, they still suffer from instability during model inversion (*cf.* Fig.1(a)) and lower fidelity in reconstructed samples (*cf.* Fig.1(b)), resulting in suboptimal performance. These issues stem from inherent flaws in GANs. Specifically, the adversarial training framework is difficult to optimize and prone to mode collapse without precise hyperparameter tuning and regularization (Brock, 2019). Moreover, GANs capture less data diversity than advanced likelihood-based models (Razavi et al., 2019; Song et al., 2021). Hence, generative models with more stable training and better mode coverage are better suited for MIA tasks.

To overcome these challenges, we propose exploring generative MIAs based on diffusion models (DMs) (Sohl-Dickstein et al., 2015; Song & Ermon, 2019; Ho et al., 2020; Song et al., 2021; Karras et al., 2022), which offer superior generative performance over GANs. However, our analysis reveals that traditional multi-step diffusion models are not directly applicable to MIAs (Sec. 3.2). This is because the iterative refinement process in reverse diffusion introduces significant computational and memory overhead, and additionally leads to the accumulation of numerical errors, resulting in inaccurate latent codes that ultimately reduce the effectiveness of the attack.

Building on recent advances in distilling diffusion models (Salimans & Ho, 2022; Luo et al., 2024; Yin et al., 2024; Zhou et al., 2024), which enable the distillation of a pretrained diffusion model into a single-step generator, we propose a novel framework for generative MIAs, termed *diffusion distillation MIAs* (DDMI). DDMI, like traditional GAN-based generative MIAs, includes two stages: ① We pretrain a multi-step diffusion model on public auxiliary datasets and apply Score identity Distillation (SiD) (Zhou et al., 2024) to create a high-performance single-step generator (*cf.* panel (b) of Fig. 2). ② This single-step generator is utilized to guide the model inversion optimization process, ensuring it remains within a meaningful image manifold (*cf.* panel (c) of Fig. 2).

Moreover, we extend generative MIA techniques, traditionally applied to classification models, to explore privacy risks in Contrastive Language–Image Pre-training (CLIP) models (Radford et al., 2021), a widely adopted multimodal model (*cf.* panel (d) of Fig. 2). Specifically, by crafting prompts like "A photo of <NAME>." and using model inversion optimization to maximize cosine similarity between text and image features, we are able to generate images that align closely with the given text prompt. This adaptation reveals the privacy vulnerabilities of multimodal models and provides insights into the extent of potential privacy leakage in such models.

Our contributions and findings are summarized as follows:

- We identify key limitations in GAN-based generative MIAs (Zhang et al., 2020; Chen et al., 2021; Nguyen et al., 2023) (Sec. 3.1) and propose a novel model inversion framework, termed the *diffusion distillation MIAs* (DDMI) (Sec. 3.3), which addresses these challenges and lays the foundation for future advancements in generative MIAs.

- To the best of our knowledge, we are the first to leverage generative MIAs to explore privacy leakage in CLIP models (Sec. 2.1), expanding the scope of MIAs to large-scale multimodal models and revealing serious privacy vulnerabilities within these models.

- Extensive experiments demonstrate that DDMI significantly outperforms SOTA GAN-based MIAs in both white-box and black-box settings (Sec. 4). These results underscore the urgent need for robust defense mechanisms to protect sensitive information in both traditional classification models and complex multimodal models like CLIP.

## 2 PROBLEM SETUP AND PRELIMINARY

### 2.1 GENERATIVE MODEL INVERSION ATTACKS

**Problem Setup.** Let the input space be $\mathcal{X} \subset \mathbb{R}^{d_\mathcal{X}}$ and the private label space be $\mathcal{Y}_{\text{pri}} = \{1, \ldots, C\}$. The *target model*, $\mathrm{M_c} \colon \mathcal{X} \to [0,1]^C$, is a classifier trained on a private dataset $\mathcal{D}_{\text{pri}}$, sampled from the private data distribution $p_{\text{pri}}(\mathbf{x}, y)$. In MIAs, for a specific class $y \in \mathcal{Y}_{\text{pri}}$, the goal is to reconstruct synthetic samples that reveal sensitive information about the private training data for that class, exploiting only access to the target model $\mathrm{M_c}$. In this setting, the adversary can query $\mathrm{M_c}$ and has general knowledge of the private data domain, but no direct access to specific details of $\mathcal{D}_{\text{pri}}$.

MIAs are typically framed as an optimization problem (Fredrikson et al., 2015), where for a given class $y$, the objective is to find a sample $\mathbf{x}$ that maximizes the likelihood of $\mathrm{M_c}$ for $y$. However, inverting DNNs trained on high-dimensional data, such as facial recognition models, is complex and often ill-posed, which can generate unrealistic features without semantic meaning. To address this challenge, Zhang et al. (2020) proposed using GANs to learn image priors that regularize the inversion process. They introduced GAN-based generative MIAs, consisting of two stages:

① **Learning Image Priors with Generative Models.** In this stage, a GAN is employed to learn image priors from public auxiliary datasets $\mathcal{D}_{\text{pub}}$, which share the same data domain as $\mathcal{D}_{\text{pri}}$, but have disjoint label spaces, *i.e.,* $\mathcal{Y}_{\text{pub}} \cap \mathcal{Y}_{\text{pri}} = \emptyset$. For brevity, we focus on approaches involving only the GAN model, following the original GAN formulation (Goodfellow et al., 2014).[1] Specifically, this process uses the generator's implicit distribution, $p_g$, to estimate the public auxiliary data distribution, $p_{\text{pub}}$. The generator, $\mathrm{G} \colon \mathcal{Z} \to \mathcal{X}$, maps input noise $\mathbf{z} \in \mathcal{Z}$ to an generated image $\mathbf{x}_g$. The discriminator, $\mathrm{D} \colon \mathcal{X} \to \mathbb{R}$, outputs the probability that an image $\mathbf{x}$ comes from $p_{\text{pub}}$ rather than $p_g$. The goal of D is accurately distinguish $\mathrm{G}(\mathbf{z})$ from $\mathbf{x}_{\text{real}}$, while the goal of G is to fool D into making mistakes. This is framed as a two-player minimax game:

$$\min_{\mathrm{G}} \max_{\mathrm{D}} \mathcal{L}_{\text{GAN}}(\mathrm{G}, \mathrm{D}) = \mathbb{E}_{\mathbf{x} \sim p_{\text{pub}}}[\log \mathrm{D}(\mathbf{x})] + \mathbb{E}_{\mathbf{z} \sim p(\mathbf{z})}[\log(1 - \mathrm{D}(\mathrm{G}(\mathbf{z})))]. \tag{1}$$

② **Model Inversion Optimization.** In the second stage, the goal is to solve an optimization problem in the well-trained generator's latent space to obtain an optimal synthetic sample, $\mathbf{x}^* = \mathrm{G}(\mathbf{z}^*)$, that resembles private data for a specific class $y$. This is formulated as:

$$\mathbf{z}^* = \arg\min_{\mathbf{z}} \mathcal{L}_{\text{id}}(\mathbf{z}; y, \mathrm{M_c}, \mathrm{G}) + \lambda \mathcal{L}_{\text{prior}}(\mathbf{z}). \tag{2}$$

Here, $\mathcal{L}_{\text{id}}(\cdot)$ denotes the the identity loss (*e.g.,* cross-entropy loss $-\log \mathbb{P}_{\mathrm{M_c}}(y|\mathrm{G}(\mathbf{z}))$), while $\mathcal{L}_{\text{prior}}(\cdot)$ serves as a regularizer on the latent code $\mathbf{z}$, with $\lambda$ controlling the balance between the two losses.

**Extend Generative MIAs to CLIP Models.** MIAs typically aim to reconstruct samples that resemble private training data in classification models, which is straightforward for models trained on

---

[1]Some MIA approaches (Chen et al., 2021; Yuan et al., 2023) utilize the target model $\mathrm{M_c}$ to generate pseudo-labels for $\mathcal{D}_{\text{pub}}$, better exploiting the private information encoded in $\mathrm{M_c}$.

simple datasets, such as frontal face images. However, CLIP models (Radford et al., 2021), trained on complex datasets containing partial/full-body images and diverse scenes with multiple objects, pose a challenge for generative MIAs, as they struggle to accurately recover training images. Therefore, for MIAs targeting CLIP models, the focus shifts to exploring potential privacy leakage and assessing how much sensitive information, such as facial features, can be inferred or reconstructed.

CLIP consists of an image encoder $M_{img}$ and a text encoder $M_{text}$ to extract features from images and text, respectively. It is trained using a contrastive loss to align corresponding image-text pairs while minimizing the similarities between unrelated pairs in the same batch. To explore privacy leakage in CLIP models, we focus on reconstructing facial images used during training. We design a prompt $\mathbf{p}$ (*e.g.,* "A photo of <NAME>.") and optimize the following objective to invert the facial data:

$$\mathbf{z}^* = \arg\min_{\mathbf{z}} \; \mathcal{L}_{id}(M_{img}(G(\mathbf{z})), M_{text}(\mathbf{p})) + \lambda\mathcal{L}_{prior}(\mathbf{z}), \tag{3}$$

where $\mathcal{L}_{id}(\cdot)$ measures the semantic similarity between image and text features, typically using cosine similarity as the default metric for comparison.

## 2.2 SCORE IDENTITY DISTILLATION

Since we employ single-step diffusion-based generators as a superior alternative to GANs for generative MIAs, this section introduces Score identity Distillation (SiD) (Zhou et al., 2024), a state-of-the-art, data-free distillation method. Unlike approaches that require access to training data, SiD relies solely on a pretrained diffusion model, making it ideal for MIAs by leveraging readily available models. SiD distills a student model $p_\theta(\mathbf{x}_g)$ from a pretrained diffusion model, enabling single-step sample generation. The generator, denoted as $G(\cdot; \boldsymbol{\theta}) \colon \mathcal{Z} \to \mathcal{X}$, is a DNN parameterized by $\boldsymbol{\theta}$ that maps noise $\mathbf{z} \sim p(\mathbf{z})$ to the generated data $\mathbf{x}_g$. Let $p_{data}(\mathbf{x}_0)$ represent the real data distribution, and $p_\theta(\mathbf{x}_g)$ the generated data distribution. Their marginals under the forward diffusion process are semi-implicit distributions (Yin & Zhou, 2018), which are intractable and expressed as follows:

$$p_{data}(\mathbf{x}_t) = \int q(\mathbf{x}_t \mid \mathbf{x}_0)p_{data}(\mathbf{x}_0) \, d\mathbf{x}_0, \quad p_{\boldsymbol{\theta}}(\mathbf{x}_t) = \int q(\mathbf{x}_t \mid \mathbf{x}_g)p_{\boldsymbol{\theta}}(\mathbf{x}_g) \, d\mathbf{x}_g,$$

where $q(\mathbf{x}_t \mid \mathbf{x}_0) = \mathcal{N}(a_t\mathbf{x}_0, \sigma_t^2\mathbf{I})$ is the forward diffusion transition kernel, with $a_t \in [0, 1]$ scaling the initial data $\mathbf{x}_0$ and $\sigma_t$ controlling the noise level added at each time step $t$.

**MESM loss.** A pretrained diffusion model provides a score network $S_\phi$, parameterized by $\phi$, which estimates the true data score at any time point $t$ of the forward diffusion process as:

$$\nabla_{\mathbf{x}_t} \ln p_{data}(\mathbf{x}_t) \approx -S_\phi(\mathbf{x}_t) := \sigma_t^{-2}(\mathbf{x}_t - a_t f_\phi(\mathbf{x}_t, t)) = \sigma_t^{-1}\boldsymbol{\epsilon}_\phi(\mathbf{x}_t),$$

where $f_\phi(\mathbf{x}_t, t)$ is the functional approximation of $\mathbb{E}[\mathbf{x}_0 \mid \mathbf{x}_t]$ and $\boldsymbol{\epsilon}_\phi(\mathbf{x}_t)$ predicts the Gaussian noise in $\mathbf{x}_t$. The model-based explicit score matching (MESM) distillation loss is defined as:

$$\mathcal{L}_{SiD}(\boldsymbol{\theta}) = \mathbb{E}_{\mathbf{x}_t \sim p_{\boldsymbol{\theta}}(\mathbf{x}_t)}[\|S_\phi(\mathbf{x}_t) - \nabla_{\mathbf{x}_t} \ln p_{\boldsymbol{\theta}}(\mathbf{x}_t)\|_2^2], \tag{4}$$

which is a type of Fisher divergence. The core concept is that if there is alignment between $p_{data}(\mathbf{x}_t)$ and $p_{\boldsymbol{\theta}}(\mathbf{x}_t)$ at any given $t$, as quantified by statistical distances like the Jensen–Shannon divergence, KL divergence, or the Fisher divergence employed here, then this implies an alignment between $p_{data}(\mathbf{x}_0)$ and $p_{\boldsymbol{\theta}}(\mathbf{x}_g)$ (Wang et al., 2023). Since $\nabla_{\mathbf{x}_t} \ln p_{\boldsymbol{\theta}}(\mathbf{x}_t)$ is unknown, we similarly assume there exists an optimal denoising network defined as: $f_{\boldsymbol{\psi}^*}(\mathbf{x}_t, t) = \mathbb{E}[\mathbf{x}_g \mid \mathbf{x}_t] = (\mathbf{x}_t + \sigma_t^2 \nabla_{\mathbf{x}_t} \ln p_{\boldsymbol{\theta}}(\mathbf{x}_t))/a_t$. Substituting this into the MESM loss in Eq. (4), the loss becomes:

$$\mathcal{L}_{SiD}(\boldsymbol{\theta}) = \mathbb{E}_{\mathbf{x}_t \sim p_{\boldsymbol{\theta}}(\mathbf{x}_t)}[\|a_t\sigma_t^{-2}(f_\phi(\mathbf{x}_t, t) - f_{\boldsymbol{\psi}^*}(\mathbf{x}_t, t)\|_2^2]. \tag{5}$$

While the MESM loss is intractable to compute analytically, SiD develops a practical solution based on three score related identities that alternates between the estimation of $\boldsymbol{\psi}^*$ and the optimization of $\boldsymbol{\theta}$, driving the student model to closely match the behavior of the pretrained diffusion model. By effectively minimizing the discrepancy between the pretrained and estimated denoising networks, SiD enables the student model to generate high-quality samples in a single step, achieving efficient performance while maintaining fidelity to the original diffusion process.

## 2.3 RELATED WORK

The problems investigated in this paper are closely related to prior research on GAN-based generative MIAs, privacy attacks on multimodal CLIP models, and diffusion model distillation approaches. A detailed introduction of these related works is deferred to Appx. A.

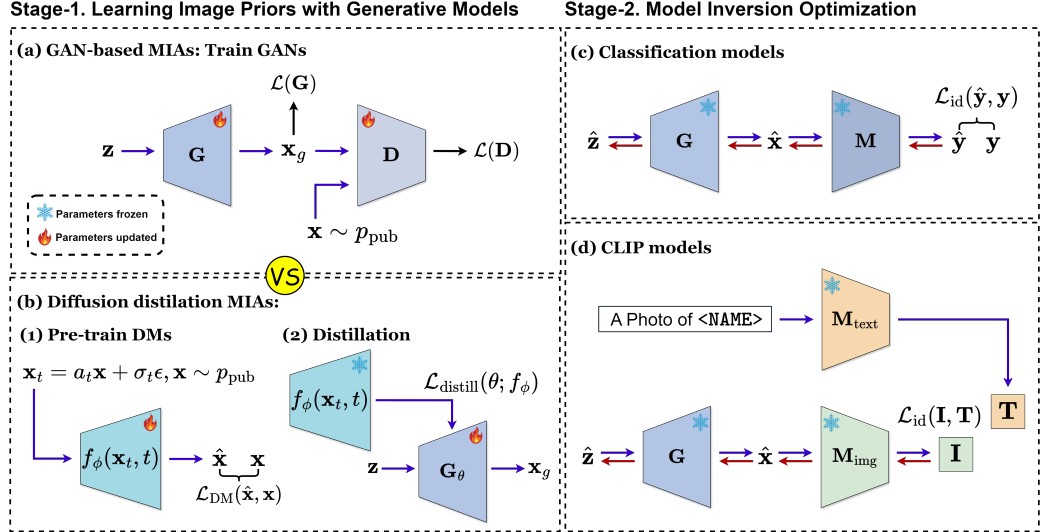

Figure 2: **Overview of traditional GAN-based MIA framework vs. *diffusion distillation MIA (DDMI)* framework.** Panel (a): In the traditional GAN-based MIAs, Stage-1 involves training a GAN model. Panel (b): In diffusion distillation MIAs, Stage-1 consists of two steps: first, pretraining a multi-step diffusion model, followed by distilling it into a single-step generator. Panel (c): Generative classifier inversion. Panel (d): Generative CLIP inversion.

## 3  SINGLE-STEP DIFFUSION MODELS FOR GENERATIVE MIAS

This section presents our novel model inversion framework, *i.e., diffusion distillation MIAs* (DDMI). First, we present and discuss the motivation behind our approach (Sec. 3.1). Next, we explain in detail why multi-step diffusion models are unsuitable for generative MIAs (Sec. 3.1). Then, we introduce the general framework of DDMI (Sec. 3.3). Finally, we apply DDMI to two types of ML models: traditional classification models and multimodal CLIP models (Sec. 3.4).

### 3.1  MOTIVATION: LIMITATIONS OF GAN-BASED GENERATIVE MIAS

While SOTA generative MIAs (Chen et al., 2021; Yuan et al., 2023; Nguyen et al., 2023) have demonstrated impressive performance in reconstructing samples that closely resemble private training data, thereby potentially exposing sensitive information, our empirical observations reveal several key limitations in these methods. The primary issues are the instability of the optimization process during model inversion (*cf.* Fig. 1(a)) and the low fidelity of the reconstructed samples (*cf.* Fig. 1(b)). This instability often prevents convergence to a desirable local optimum, resulting in suboptimal inversion performance. Additionally, the low fidelity of the reconstructed samples limits their ability to accurately capture fine-grained sensitive features from the private training data.

We attribute these limitations to inherent flaws in GANs. Specifically, the two-player minimax game outlined in Eq. (1) is notoriously difficult to optimize and prone to collapse without careful selection of hyperparameters and regularization techniques (Miyato et al., 2018; Brock, 2019). Moreover, GANs struggle to capture data diversity as effectively as SOTA likelihood-based models (Razavi et al., 2019; Song et al., 2021), making them less suitable for generative MIAs. Consequently, generative models with more stable training procedures and better mode coverage are more appropriate for MIA tasks. To address these issues, we propose exploring generative MIAs based on diffusion models (Ho et al., 2020; Karras et al., 2022), which offer superior generative performance compared to GANs. In particular, diffusion models offer more stable training and better capture data diversity.

### 3.2  WHY MULTI-STEP DIFFUSION MODELS FALL SHORT FOR GENERATIVE MIAS

Although generative MIAs based on diffusion models offer a promising approach, directly applying multi-step diffusion models presents significant challenges associated with the image generation (denoted as Process (6a)) and latent code update processes (Process (6b)), as outlined below:

$$\texttt{Image Generation } (\mathcal{Z} \to \mathcal{X}) \texttt{: } \mathbf{z} = \mathbf{x}_T \xrightarrow{f_\phi(\mathbf{x}_T, T)} \mathbf{x}_{T-1} \to \cdots \xrightarrow{f_\phi(\mathbf{x}_1, 1)} \mathbf{x}_0, \quad (6a)$$

$$\texttt{Latent Code Update } (\mathcal{X} \to \mathcal{Z}) \texttt{: } \mathbf{z} = \mathbf{x}_T \xleftarrow{\frac{\partial \mathbf{x}_{T-1}}{\partial \mathbf{x}_T}} \mathbf{x}_{T-1} \leftarrow \cdots \xleftarrow{\frac{\partial \mathbf{x}_0}{\partial \mathbf{x}_1}} \mathbf{x}_0, \quad (6b)$$

**Challenge-1: High Computational and Memory Overhead.** In diffusion models, Process (6a) involves solving an ordinary differential equation (ODE) numerically. This process gradually denoises an initial random noise $\mathbf{z}$ to generate a coherent and high-quality sample $\mathbf{x}_0$. However, this procedure is computationally intensive, requiring a considerable number of function evaluations (NFEs). For instance, in EDM (Karras et al., 2022), generating a $64 \times 64$ FFHQ image requires 79 NFEs to achieve a good performance. Additionally, in Process (6b), gradient backpropagation requires storing derivatives at each step. For example, using the commonly employed Euler's method, one need store $\frac{\partial \mathbf{x}_t}{\partial \mathbf{x}_{t+1}} \propto \frac{\partial f(\mathbf{x}_t; t)}{\partial \mathbf{x}_{t+1}}$ ($t = 0, \cdots, T-1$), resulting in significant memory overhead.

**Challenge-2: Accumulation of Numerical Errors Leading to Inaccurate Latent Code.** In Process (6a), each step of the numerical solver introduces truncation errors that accumulate over the course of $T$ steps. Furthermore, these errors can propagate through Process (6b), causing inaccuracies in the correspondence between the latent code and the image. Consequently, this can lead to a suboptimal or inaccurate latent code during the update, potentially degrading inversion performance.

### 3.3 LEVERAGING DISTILLED SINGLE-STEP DIFFUSION MODELS FOR MODEL INVERSION

Building on recent advancements in the distillation of diffusion models (Salimans & Ho, 2022; Luo et al., 2024; Yin et al., 2024; Zhou et al., 2024), which enable the compression of knowledge from a pretrained diffusion model into a more efficient single-step generator, we propose a novel framework for generative MIAs. This framework, termed *diffusion distillation MIAs* (DDMI), follows a similar structure to traditional GAN-based generative MIAs, with two primary stages:

① **Learning Image Priors with Diffusion Models.**

This stage consists of two critical steps:

- **Step-1: Pretrain a Multi-Step Diffusion Model.** Initially, a multi-step diffusion model is pretrained using public auxiliary datasets $\mathcal{D}_{\text{pub}}$, which learns to generate high-quality image samples by reversing the noise corruption process, following a denoising objective:

$$\min_\phi \mathcal{L}_{\text{DM}}(\phi) = \mathbb{E}_{\mathbf{x} \sim p_{\text{pub}}, \, \mathbf{n} \sim \mathcal{N}(\mathbf{0}, \sigma_t^2 \mathbf{I}), \, t} \|f_\phi(a_t \mathbf{x} + \mathbf{n}, t) - \mathbf{x}\|_2^2. \quad (7)$$

- **Step-2: Distill the Multi-Step Diffusion Model into a Single-Step Generator.** After pretraining, the multi-step diffusion model $f_\phi$ is distilled into a single-step generator, denoted as $G_\theta$. This is achieved by minimizing a distillation loss function as follows:

$$\min_\theta \mathcal{L}_{\text{distill}}(\theta; f_\phi). \quad (8)$$

  This distillation process preserves learned image priors while significantly reducing computational complexity. In the context of inverting classifiers, to fully exploit the private information encoded within the target model $M_c$, we can generate pseudo-labels to the public dataset $\mathcal{D}_{\text{pub}}$ using $M_c$. Step-1 then becomes training a conditional diffusion model $f_\phi(\cdot; y)$. In Step-2, an *inversion-specific* distillation process can be designed as follows: $\min_\theta \mathcal{L}_{\text{distill}}(\theta; f_\phi) + \mathcal{L}_{\text{id}}(\mathbf{z}; y, M_c, G_\theta(\cdot; y))$. In practice, we first apply SiD, then perform further distillation with the identity loss to finalize the process.

② **Model Inversion Optimization with the Distilled Generator.**

With the distilled single-step generator $G_\theta$, we perform model inversion. Given a well-trained target model $M$, we conduct model inversion by constraining the optimization process to the latent space of the distilled generator. The optimization objective is formulated as follows:

$$\mathbf{z}^* = \arg\min_{\mathbf{z}} \ \mathcal{L}_{\text{id}}(\mathbf{z}; M, G_\theta) + \lambda \mathcal{L}_{\text{prior}}(\mathbf{z}), \quad (9)$$

This approach allows for effective recovery of input data with greater computational efficiency, leveraging the distilled knowledge of the generator.

### 3.4 APPLICATIONS OF DDMI: CLASSIFICATION AND MULTIMODAL MODELS

In our implementation, we utilize Score identity Distillation (SiD), a state-of-the-art approach that enables a single-step generator $G_{\theta}$ to potentially surpass the performance of the original pretrained diffusion model $f_{\phi}$. While the SiD loss (Eq. (5)) is a theoretical formulation that cannot be estimated analytically, we employ an approximation to optimize over the latent code $\mathbf{z}$ as the prior loss, ensuring that $G_{\theta}(\mathbf{z})$ remains within the image manifold learned by $f_{\phi}$. We apply DDMI to both traditional classification models and multimodal CLIP models, as detailed below.

**Diffusion distillation MIAs on classification models.** When the target model $M$ represents a standard classification model $M_c$, Eq. (9) is instantiated as follows:

$$\mathbf{z}^* = \arg \min_{\mathbf{z}} \ \mathcal{L}_{\mathrm{id}}(\mathbf{z}; y, M_c, G_{\theta}) + \lambda \mathcal{L}_{\mathrm{SiD}}(\mathbf{z}), \tag{10}$$

where $\mathcal{L}_{\mathrm{id}}(\cdot)$ can be implemented with various losses, including cross-entropy loss (Zhang et al., 2020; Chen et al., 2021), Poincaré loss (Struppek et al., 2022), max-margin loss (Yuan et al., 2023), or logit maximization loss (Nguyen et al., 2023), as used in existing literature.

**Diffusion distillation MIAs on CLIP models.** When $M$ represents a multimodal CLIP model that consists of an image encoder $M_{\mathrm{img}}$ and a text encoder $M_{\mathrm{text}}$, Eq. (9) is expressed as:

$$\mathbf{z}^* = \arg \min_{\mathbf{z}} \ \mathcal{L}_{\mathrm{id}}(M_{\mathrm{img}}(G_{\theta}(\mathbf{z})), M_{\mathrm{text}}(\mathbf{p})) + \lambda \mathcal{L}_{\mathrm{SiD}}(\mathbf{z}), \tag{11}$$

where $\mathcal{L}_{\mathrm{id}}(\cdot)$ is implemented as cosine similarity (Radford et al., 2021), which measures the semantic similarity between image and text features. Algorithmic details of DDMI are deferred to Appx. B.

## 4 EXPERIMENTS

We compare single-step diffusion model (SDM)-based MIAs (*i.e.,* DDMI) with SOTA GAN-based methods for classifier inversion on real-world facial recognition tasks, specifically targeting *low-resolution* ($64 \times 64$) images. Despite promising results from traditional metrics, these attacks often suffer from poor visual fidelity in the reconstructions. Our evaluation includes GMI (Zhang et al., 2020), LOMMA (Nguyen et al., 2023), and PLG-MI (Yuan et al., 2023) in the white-box setting, as well as BREP-MI (Kahla et al., 2022) in the black-box setting. For CLIP inversion, the objective is to infer and reconstruct facial images in large-scale training data. In this initial exploration of *generative* CLIP inversion, we use SDM and pretrained StyleGAN (Karras et al., 2020) as image priors, compared to the input space optimization baseline, CLIPInversion (Kazemi et al., 2024).

### 4.1 EXPERIMENTAL SETUP

This section briefly introduces the experimental setups. For further details, please refer to Appx. C.

**Datasets and models.** For classifier inversion, we follow the standard MIA literature, utilizing the CelebA (Liu et al., 2015), and FFHQ (Karras et al., 2019) datasets. We train VGG16 (Simonyan & Zisserman, 2015) and face.evoLVe (Wang et al., 2021b) as target models. For CLIP inversion, we evaluate using the FaceScrub dataset (Ng & Winkler, 2014), which is a subset of the LAION-400M dataset (Schuhmann et al., 2021), used to train open-source CLIP models. We perform model inversion on multiple CLIP models using three image feature extractors: ViT-B/16, ViT-B/32, and ViT-L/14 (Dosovitskiy et al., 2021). The text encoder architecture for all CLIP models remains consistent with the original CLIP paper. Training details of these models are provided in Appx. C.2. A summary of attack methods, target models, and datasets used is shown in Tab. 4.

**Evaluation Metrics.** To evaluate the performance of MIAs, we assess whether the reconstructed images reveal private information about the target identity. Following prior work, we conducted both qualitative and quantitative evaluations. For qualitative analysis, we visually inspected the reconstructed images. For quantitative evaluation, we adopted the following metrics from the literature (Zhang et al., 2020), including top-1 accuracy (Acc@1), top-5 accuracy (Acc@5), K-Nearest Neighbors Distance (KNN Dist), and Fréchet inception distance (FID). Further details on these and additional metrics are provided in Appx. C.4, with attack parameters detailed in Appx. C.5.

Table 1: Comparison with GMI and LOMMA (GMI) in low-resolution tasks. $\mathcal{D}_{\text{pri}}$ = CelebA, generative models are trained on $\mathcal{D}_{\text{pub}}$ = CelebA or FFHQ. The symbol ↓ (or ↑) indicates smaller (or larger) values are preferred, and the green numbers represent the performance improvement.

| Target Model | Method | CelebA | | | | FFHQ | | | |
|---|---|---|---|---|---|---|---|---|---|
| | | Acc@1↑ | Acc@5↑ | KNN Dist ↓ | FID↓ | Acc@1↑ | Acc@5↑ | KNN Dist ↓ | FID↓ |
| VGG16 | GMI | 18.28 | 39.13 | 1717.40 | 53.04 | 9.08 | 24.07 | 1806.10 | 43.06 |
| | w/ SDM (ours) | $21.85_{\uparrow 3.57}$ | $43.77_{\downarrow 4.64}$ | $1674.15_{\downarrow 43.25}$ | $41.51_{\downarrow 11.53}$ | $13.70_{\uparrow 4.62}$ | $31.45_{\uparrow 7.38}$ | $1736.01_{\downarrow 70.09}$ | $42.83_{\downarrow 0.23}$ |
| | LOMMA (GMI) | 73.72 | 92.21 | 1316.63 | 48.87 | 54.39 | 79.44 | 1437.09 | 38.40 |
| | w/ SDM (ours) | $82.97_{\uparrow 9.25}$ | $94.39_{\uparrow 2.18}$ | $1233.82_{\downarrow 82.81}$ | $25.78_{\downarrow 23.09}$ | $65.19_{\uparrow 10.80}$ | $87.10_{\uparrow 7.66}$ | $1391.85_{\downarrow 45.24}$ | $34.00_{\downarrow 4.40}$ |
| face.evoLVe | GMI | 26.39 | 50.97 | 1645.10 | 54.48 | 12.46 | 30.08 | 1772.59 | 45.90 |
| | w/ SDM (ours) | $27.89_{\uparrow 1.50}$ | $51.62_{\uparrow 0.65}$ | $1629.20_{\downarrow 15.90}$ | $41.15_{\downarrow 13.33}$ | $17.24_{\uparrow 4.78}$ | $36.51_{\uparrow 6.43}$ | $1711.84_{\downarrow 60.75}$ | $45.32_{\downarrow 0.58}$ |
| | LOMMA (GMI) | 80.21 | 94.79 | 1270.79 | 50.38 | 62.77 | 85.15 | 1406.75 | 42.47 |
| | w/ SDM (ours) | $86.71_{\uparrow 6.50}$ | $95.14_{\uparrow 0.35}$ | $1209.17_{\downarrow 61.62}$ | $25.93_{\downarrow 24.45}$ | $71.23_{\uparrow 8.46}$ | $90.17_{\uparrow 5.02}$ | $1368.14_{\downarrow 38.61}$ | $35.04_{\downarrow 7.43}$ |

## 4.2 MAIN RESULTS

Detailed evaluations for classifier inversion and CLIP inversion are provided in Secs. 4.2.1 and 4.2.2.

### 4.2.1 CLASSIFIER INVERSION

In the main experiments, our approach (*i.e.,* w/ SDM) refers to replacing the GAN-based MIA framework with the SDM-based framework, while keeping the identity loss unchanged during model inversion. Additional results, including evaluations against SOTA inversion defenses, comparisons in the black-box setting, and experiments with additional evaluation metrics, are provided in Appx. D.

**Comparison with GAN-based MIAs without using the target model in prior learning.** For each baseline setup, we present results using SDM. As shown in Tab. 1, DDMI consistently outperforms baseline white-box attacks, validating its effectiveness. Specifically, integrating SDM with the baseline setup significantly improves attack accuracy. The KNN feature distance further confirms that our method reconstructs samples that closely resemble the private training data. Additionally, qualitative results demonstrate a substantial reduction in FID, indicating improved visual quality in the reconstructed samples. Despite the distribution shift between the private training dataset (CelebA) and the public auxiliary dataset (FFHQ), our method maintains strong inversion performance. Qualitative examples of the reconstructed samples are presented in Figs. 5 and 6 in Appx. D.3.

**Comparison with GAN-based MIAs incorporating the target model in prior learning (*i.e.,* PLG-MI).** In this setup, we combine the diffusion distillation loss with the identity loss, following the inversion-specific distillation approach outlined in Sec. 3.3, which enables the model to learn identity-wise priors. The results, presented in Tab. 3, demonstrate that our inversion-specific method outperforms SOTA PLG-MI in terms of attack accuracy and KNN distance metrics, as well as in FID score. Qualitative examples of reconstructed samples are shown in Fig. 7 in Appx D.3.

Despite strong metrics, visual inspection reveals that the reconstructed samples do not fully align with the high scores of both methods, particularly the baseline. We hypothesize that methods relying on the target model for pseudo-labeling public datasets depend heavily on its discriminative power. A more robust target model could improve visual quality by better identifying public samples resembling private data. This suggests that traditional metrics, especially attack accuracy, may not fully capture inversion success in some cases, highlighting the need for more reliable evaluation methods.

### 4.2.2 CLIP INVERSION

For CLIP model inversion, our approach leverages image priors learned by SDM or pretrained Style-GAN models. We compare this approach with CLIPInversion, introduced by (Kazemi et al., 2024), which utilizes direct input space optimization for inversion. To evaluate the effectiveness of our method, we perform inversion using the simple prompt "A photo of <NAME>." The effect of more detailed prompt designs on the inversion performance is reserved for the ablation study.

**Comparison with CLIPInversion baseline.** The results in Tab. 2 show that our approach outperforms the baseline by reconstructing semantically meaningful images (top panel of Fig. 3), primarily due to the integration of generative priors during inversion optimization. However, inversion per-

Table 2: Model inversion performance comparison on CLIP models with different image encoders. $\mathcal{D}_{pri}$ = FaceScrub (a subset of LAION-400M), generative models are trained on $\mathcal{D}_{pub}$ = FFHQ.

| Target Model | Method | Acc@1↑ | Acc@5↑ | KNN Dist↓ |
|---|---|---|---|---|
| **ViT-B/32** | CLIPInversion | 5.94 | 20.63 | 1.0058 |
| | w/ SDM | 7.35 | 20.65 | 0.9596 |
| | w/ StyleGAN | 6.50 | 19.80 | 0.9727 |
| **ViT-B/16** | CLIPInversion | 2.50 | 13.13 | 0.9962 |
| | w/ SDM | 6.65 | 18.80 | 0.9536 |
| | w/ StyleGAN | 8.70 | 22.60 | 0.9146 |
| **ViT-L/14** | CLIPInversion | 5.63 | 10.94 | 1.1018 |
| | w/ SDM | 6.45 | 16.65 | 0.9099 |
| | w/ StyleGAN | 7.95 | 21.05 | 0.8829 |

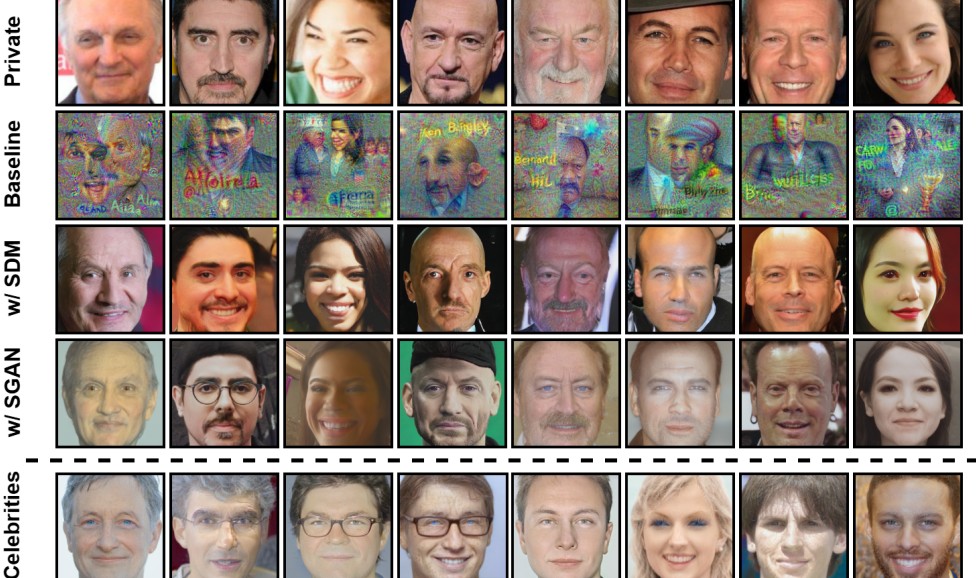

Figure 3: **Visualization of reconstructed samples from CLIPInversion and generative CLIP inversion on the ViT-L/14 image encoder.** Top: The first row shows private images, the second row displays images reconstructed by baseline, and the third and fourth rows present images reconstructed using generative inversion via SDM and StyleGAN (referred to as SGAN). Bottom: Reconstructed images of well-known figures, from left to right: Geoffrey Hinton, Yoshua Bengio, Yann LeCun, Bill Gates, Elon Musk, Taylor Swift, Lionel Messi, and Stephen Curry.

formance on CLIP models, based on traditional metrics, is notably lower compared to classifier inversion settings. Nonetheless, our experiments reveal a crucial phenomenon: as model capability increases, the risk of privacy leakage also rises. This trend aligns with findings in classifier inversion, where similar conclusions have been observed and theoretically proven. Specifically, models with stronger predictive power tend to be more vulnerable to inversion attacks (Zhang et al., 2020).

As for the suboptimal inversion performance, we hypothesize it is due to the relatively low presence of FaceScrub celebrity images in the CLIP's large-scale training dataset. Therefore, CLIP models may not effectively learn their discriminative features. To further explore potential privacy risks, we reconstructed images of more well-known celebrities, assuming their higher frequency in CLIP's training data. The examples are shown in bottom panel of Fig. 3, demonstrating that these reconstructions closely resemble the celebrities, highlighting privacy vulnerabilities in CLIP models.

Table 3: Comparison with PLG-MI in low-resolution tasks. $\mathcal{D}_{\text{pri}}$ = CelebA, conditional generative models are trained on $\mathcal{D}_{\text{pub}}$ = FFHQ.

| Target Model | Method | Acc@1↑ | KNN Dist↓ | FID↓ |
|---|---|---|---|---|
| **VGG16** | PLG-MI | 86.37 | 1283.21 | 37.30 |
| | DDMI | 88.37 | 1261.57 | 26.41 |
| **face.evoLVe** | PLG-MI | 93.83 | 1210.88 | 34.59 |
| | DDMI | 95.56 | 1184.36 | 30.80 |

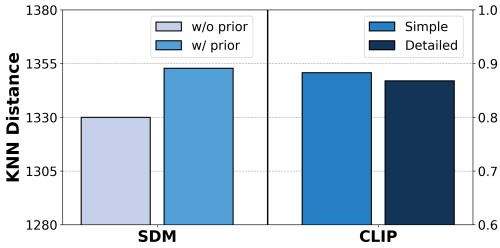

Figure 4: **Ablation study**. Left: Impact of prior loss. Right: Impact of prompt detail level.

In future work, we plan to expand the dataset by collecting images of globally recognized celebrities, extending beyond the existing FaceScrub dataset. Since images of these celebrities are more likely to appear frequently in training datasets, this expansion would provide a more diverse and representative foundation for evaluation. This approach aims to enhance the accuracy of our assessments and enable a deeper investigation into the privacy leakage risks posed by CLIP models.

**Comparison between SDM-based and StyleGAN-based generative CLIP inversion.** For image priors, we used a pretrained StyleGAN model on the $1024 \times 1024$ FFHQ dataset and a single-step diffusion generator distilled from a $256 \times 256$ latent diffusion model trained on the same dataset. The SDM-based method showed worse inversion performance both quantitatively and qualitatively compared to the StyleGAN-based one. Because diffusion models struggle to match StyleGAN's generative performance at $256 \times 256$ or higher resolutions of FFHQ (Dao et al., 2023). In our experiments, the FID for the SDM is 3.85, while the FID for the pretrained StyleGAN is 2.84.

### 4.3 ABLATION STUDY

In this section, we present ablation studies to further investigate DDMI's performance in both classifier inversion and CLIP inversion tasks. Further discussions are available in Appx. E.

**Prior loss.** We examine the effect of applying or omitting the prior loss on DDMI's inversion performance. The prior loss is designed to constrain the reconstructed samples within the learned image manifold. As shown in the left panel of Fig. 4, adding the prior loss increases the KNN feature distance. This occurs because the label sets of the public and private datasets do not overlap, meaning the private data is likely located in low-density regions of the public data distribution. The prior loss tends to constrain the reconstructed samples to the high-density regions of the public data distribution to improve visual quality, which in turn affects the model inversion performance.

**More detailed templates.** The prompts used in CLIP inversion are critical, as detailed and structured text prompts enable the text encoder to generate more accurate features of the target. These text features directly influence the image optimization process. We used a more complex prompt: "A close-up portrait of `<Name>` with `<hair>`, `<eye>`, `<lip>`, and `<face shape>`, emphasizing their `<facial characteristic>` and `<appearance>`." We averaged the KNN distance for 40 celebrities from the FaceScrub dataset, and the results are presented in the right panel of Fig. 4, indicating that more detailed prompts can improve inversion performance for MIAs on CLIP.

### 5 CONCLUSION

In this research, we identified key limitations in GAN-based generative MIAs and addressed them by proposing a novel framework based on single-step diffusion models, which we term *diffusion distillation MIAs* (DDMI). Notably, to the best of our knowledge, this is also the first study to investigate generative model inversion attacks on CLIP models, a previously uninvestigated area. Through extensive experimentation, we demonstrate that our approach not only improves traditional performance metrics but also significantly enhances the visual fidelity of reconstructed samples. Our findings further highlight the urgent need for robust and effective privacy-preserving mechanisms during model training, particularly for CLIP models, a largely underexplored area.

ETHICS STATEMENT

In this paper, we introduce a novel model inversion framework, DDMI, to enhance the effectiveness of generative MIAs. Additionally, we extend generative MIAs to CLIP models, offering significant advancements and new opportunities for future research. From a societal perspective, our work sheds light on critical privacy risks in machine learning models that could potentially expose sensitive training data if exploited. By identifying these vulnerabilities, we seek to raise awareness and drive the development of robust defense strategies and privacy-preserving techniques, which are crucial for safeguarding machine learning systems. While there is a risk of misuse, the overall benefit of increasing awareness and reinforcing security measures far outweighs these concerns.

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

# Appendix

C ONTENTS

## A   RELATED WORK

**Model Inversion Attacks on Classification Models.** Zhang et al. (2020) introduced GAN-based generative MIAs, significantly advancing the field. Recent innovations build upon this framework, with some focusing on GAN training. For instance, Chen et al. (2021) developed an inversion-specific GAN, while Yuan et al. (2023) proposed pseudo label-guided MIAs using conditional GANs to more effectively extract knowledge from public datasets. Other works have focused on improving optimization techniques, such as Wang et al. (2021a); Struppek et al. (2022); Kahla et al. (2022), which developed advanced inversion loss functions, and Han et al. (2023); An et al. (2022); Kahla et al. (2022), which explored alternative gradient-based strategies for black-box settings.

**Inference Attacks on CLIP Models.** The CLIP model advances multimodal learning by integrating image and text encoders into a shared embedding space for semantic similarity (Radford et al., 2021). Hintersdorf et al. (2024) explored personal membership leakage by querying the model with both images and text. Ko et al. (2023) used cosine similarity between visual and textual features for membership inference under a weak supervision framework, while Li et al. (2024) extend membership inference attack using only text queries. Kazemi et al. (2024) examined CLIP's ability to blend concepts and detect potential gender biases. In this paper, In this paper, we leverage generative MIAs to investigate privacy leakage issues of CLIP models.

**Distilled Diffusion Models.** Diffusion models have received significant attention for generating high-quality images through iterative noise refinement, though their multi-step process can be computationally inefficient. To address this limitation, researchers have explored methods to distill the reverse diffusion chain into more efficient processes. Salimans & Ho (2022) pioneered the concept of progressive distillation, aiming to reduce the number of steps without sacrificing generation quality. Song et al. (2023) introduced consistency models, improving output consistency along the ODE trajectory. Recent advancements (Luo et al., 2024; Yin et al., 2024; Zhou et al., 2024) have successfully distilled single-step generators, significantly reducing generation costs while often approaching or even surpassing the performance of the original pretrained diffusion models.

## B   THE ALGORITHMIC REALIZATION OF SDM-MI

This section provides the algorithmic implementation of DDMI (*cf.* Alg. 1), particularly focusing on its application to both traditional classification and multimodal CLIP models.

## C   EXPERIMENTAL DETAILS

### C.1   HARD- AND SOFTWARE DETAILS

For MIAs focusing on low-resolution facial recognition tasks, we conducted experiments on Oracle Linux Server 8.10, equipped with NVIDIA Tesla V100-PCIE-32GB GPUs. The configuration included CUDA 11.6, Python 3.8.0, and PyTorch 1.12.1. In our experiments with MIAs targeting multimodal CLIP models, we experimented on on the same server configuration, leveraging NVIDIA Ampere A100-80G GPUs with CUDA 11.7, Python 3.9.18, and PyTorch 1.13.1.

### C.2   TARGET MODELS

To train the target models on CelebA images with a resolution of $64 \times 64$, we used the training script provided at https://github.com/sutd-visual-computing-group/Re-thinking_MI. These models were trained for 50 epochs using the SGD optimizer, with an initial learning rate of $10^{-2}$, a momentum of 0.9, and a weight decay of $10^{-4}$. The batch size was set to 64. The learning rate decay schedule is model-specific; please refer to the script for detailed information. The CLIP models used in our experiments are pretrained models on the 400 million-image LAION-400M dataset.

---

**Algorithm 1** Diffusion Distillation Model Inversion Attacks

---

**Input:** Target model M, public auxiliary dataset $\mathcal{D}_{\text{pub}}$, and the set of identity set to be reconstructed $C$.

```
 1: # Stage-1.  Learning Image Priors with Diffusion Models
```
 2: **Pretrain** a multi-step diffusion model $f_\phi$ using Eq. (7);
 3: **Distill** the multi-step diffusion model $f_\phi$ into a single-step generator $G_\theta$, using Eq. (8);
```
 4: # Stage-2.  Model Inversion Optimization with the Distilled
    Generator
 5: reconstructed_samples = [];
```
 6: **if** M is a classifier **then**
 7:    **for** each target identity $y$ in $C$ **do**
 8:       **Initialize** latent codes: $\mathbf{Z} = \{\mathbf{z}_i \mid \mathbf{z}_i \in \mathcal{Z}, i = 1, \ldots, N\}$;
 9:       **Obtain** optimized latent codes $\hat{\mathbf{Z}}$ using Eq. (9);
10:       **Generate** reconstructed samples: $\mathcal{D}_{\text{rec}} = \{\hat{\mathbf{x}} = G_\theta(\hat{\mathbf{z}}) \mid \hat{\mathbf{z}} \in \hat{\mathbf{Z}}\}$;
```
11:       reconstructed_samples += 𝒟_rec;
```
12:    **end for**
13: **else if** M is a CLIP model **then**
14:    **for** each target identity $y$ in $C$ **do**
15:       **Design** a prompt $\mathbf{p}$ with target identity $y$;
16:       **Initialize** latent codes: $\mathbf{Z} = \{\mathbf{z}_i \mid \mathbf{z}_i \in \mathcal{Z}, i = 1, \ldots, N\}$;
17:       **Obtain** optimized latent codes $\hat{\mathbf{Z}}$ using Eq. (11);
18:       **Generate** reconstructed samples: $\mathcal{D}_{\text{rec}} = \{\hat{\mathbf{x}} = G_\theta(\hat{\mathbf{z}}) \mid \hat{\mathbf{z}} \in \hat{\mathbf{Z}}\}$;
```
19:       reconstructed_samples += 𝒟_rec;
```
20:    **end for**
21: **end if**
```
22: Output: reconstructed_samples.
```

---

## C.3   Evaluation Models

In the low-resolution classifier inversion setting, where classifiers are trained on the $64 \times 64$ resolution CelebA dataset, we use an evaluation model available for download at `https://github.com/sutd-visual-computing-group/Re-thinking_MI`. This model is derived from the face.evoLVe model (Wang et al., 2021b), incorporating a modified ResNet-50 backbone, and achieves a reported test accuracy of $95.88\%$. For detailed information regarding the training process and implementation specifics, please refer to Zhang et al. (2020).

For CLIP inversion, we train Inception-v3 evaluation models following the code and guidelines available at `https://github.com/LukasStruppek/Plug-and-Play-Attacks`. For specific training details, please refer to (Struppek et al., 2022). These models achieve test accuracy of $96.53\%$ on the FaceScrub dataset. Additionally, we utilize the pretrained FaceNet model (Schroff et al., 2015), available at `https://github.com/timesler/facenet-pytorch`, to compute K-nearest neighbors distances, which provide a measure of similarity between training samples and reconstructed samples in the facial recognition tasks.

## C.4   Evaluation Metrics

**Attack Accuracy (Attack Acc).** Following previous work (Zhang et al., 2020), we use an evaluation model (typically one with better generalization ability than the target model) trained on the same dataset to assess the reconstructed images (see Tab. 4). This model serves as a proxy for human judgment. Attack accuracy is measured by the percentage of predictions that correctly match the target identity, with top-1 (Acc@1) and top-5 (Acc@5) accuracy as the primary metrics.

**K-Nearest Neighbors Distance (KNN Dist).** KNN Distance measures the $l_2$ distance between the reconstructed images and their nearest neighbors in the target model's training data within the embedding space. This metric reflects the visual similarity between the reconstructed images and the original training data. For MIAs targeting low-resolution tasks, we compute KNN Dist using the penultimate layer of the evaluation model. In the context of CLIP inversion, we utilize the penulti-

Table 4: A summary of experimental setups.

| Scenario | Type | MIAs | Private Dataset | Public Dataset | Target Model | Evaluation Model |
|---|---|---|---|---|---|---|
| **Classification** | White-box | GMI / LOMMA | CelebA | CelebA / FFHQ | VGG16 / face.evoLVe | face.evoLVe |
| | Label-only | BREP-MI | CelebA | CelebA | VGG16 | face.evoLVe |
| **Multimodal** | White-box | CLIPInversion | FaceScrub / (LAION-400M) | FFHQ | ViT-B/32 ViT-B/16 ViT-L/14 | Pretrained FaceNet |

mate layer of a pretrained FaceNet (Schroff et al., 2015), with A smaller KNN distance indicates a higher degree of similarity between the reconstructed images and the training set.

**Fréchet Inception Distance (FID).** FID is a standard metric for evaluating the similarity between generated and real data distributions. It calculates the distance between the feature representations of real and generated images using an Inception network. We compute the FID score between all reconstructed images and the private dataset. A lower FID score indicates a closer alignment between the two distributions, reflecting higher visual quality and realism in the generated samples.

**Precision & Recall (Kynkäänniemi et al., 2019), Density & Coverage (Naeem et al., 2020).** We calculate these four metrics on a per-class basis to assess sample diversity. While diversity is not crucial for MIA success, as the adversary's primary goal may not be to capture the full range of private data, it still provides useful reference points:

- **Precision** measures the proportion of generated samples that are realistic and closely match the distribution of real data. In other words, it quantitatively assesses the quality of the generated images by evaluating how closely they resemble the true data distribution.

- **Recall** evaluates the generative model's ability to cover the entire data distribution. It comprehensively measures how well the model captures the full diversity of the real data, ensuring that it generates a wide variety of samples from the target distribution.

- **Density** refines the concept of precision by measuring how closely generated samples cluster around real data. A higher density score indicates that the generated samples are more tightly clustered around real data points, reflecting their closeness in feature space.

- **Coverage** complements density by measuring how well generated samples span the real data distribution, indicating the model's ability to capture data diversity, particularly whether it generates samples from underrepresented regions of the distribution.

## C.5 ATTACK PARAMETERS

For the white-box attacks, we inverted 100 identities, generating 100 reconstructed samples for each identity. In the experiments for GMI and LOMMA (GMI), model inversion optimization was run for $1,000$ iterations for the baseline and 300 iterations for DDMI. For the PLG-MI experiments, model inversion optimization ran for 100 iterations for both the baseline and DDMI.

For the label-only attack (BREP-MI), we generated 10 samples per identity for 50 identities from the CelebA dataset, with the number of sampling points set to 64. During optimization, the initial radius was set to 2, with an expansion coefficient of 1.2 to explore a wider range of radius values. The maximum number of optimization steps was set to $1,000$.

For attacks on CLIP, we generated 20 samples per experiment across 100 identities from the FaceScrub dataset. For the baseline CLIPInversion, the samples were optimized over $2,000$ iterations with a learning rate of 0.1. The image size started at $64 \times 64$, scaling up at the 900-th and $1,800$-th iterations. All other hyperparameters were kept at their default values. In our generative CLIPInversion setup, we utilized a $1,024 \times 1,024$ pretrained StyleGAN model on the FFHQ dataset for the StyleGAN-based approach, and a $256 \times 256$ single-step diffusion generator distilled from a pre-

Table 5: Comparison with BREP-MI (black-box setting) in low-resolution tasks. The target model M = VGG16 is trained on $\mathcal{D}_{\text{pri}}$ = CelebA. Generative models are trained on $\mathcal{D}_{\text{pub}}$ = CelebA or FFHQ. The symbol ↓ (or ↑) indicates smaller (or larger) values are preferred, and green numbers represent improvements with SDM.

| Method | CelebA | | | | FFHQ | | | |
|---|---|---|---|---|---|---|---|---|
| | Acc@1↑ | Acc@5↑ | KNN Dist↓ | FID↓ | Acc@1↑ | Acc@5↑ | KNN Dist↓ | FID↓ |
| BREP-MI | 48.00 | 78.00 | 1128.53 | 54.15 | 24.00 | 53.00 | 1269.80 | 56.47 |
| w/ SDM (ours) | 53.00$_{\uparrow 5.00}$ | 81.00$_{\uparrow 3.00}$ | 1015.83$_{\downarrow 112.70}$ | 47.55$_{\downarrow 6.60}$ | 25.00$_{\uparrow 1.00}$ | 55.00$_{\uparrow 2.00}$ | 1194.68$_{\downarrow 75.12}$ | 54.48$_{\downarrow 1.99}$ |

trained latent diffusion model. We optimized for 300 iterations, resizing the generated images to $224 \times 224$ for evaluation.

### C.6 EXPERIMENTAL DETAILS FOR FIG. 1

We present the experiment details for generating Fig. 1. In the motivation experiments, we detail the experimental setup used to generate the presented results. In our motivation experiments, we investigate two primary challenges of SOTA GAN-based MIAs: unstable optimization and low image fidelity. To examine the unstable optimization issue, we visualize the top-1 attack accuracy as the number of optimization iterations increases. To assess the limitation of low image fidelity, we visualize the fidelity of generated images measured by KNN distance. Unless otherwise specified, all other hyperparameters for each attack setup remain consistent with those outlined in Appx. C.5.

In Fig. 1(a), we compare the top-1 attack accuracy of our SDM-based LOMMA (GMI) with LOMMA (GMI), and KEDMI as the number of optimization iterations increases in the low-resolution setting. For all attack setups, the target model is VGG16 trained on the CelebA dataset, and we use the FFHQ dataset to learn prior knowledge. The evaluation model employed is face.evoLve also trained on the CelebA dataset. We optimize 400 iterations in the SDM-based setup and 1,000 iterations in the other setups. Attack accuracy is recorded every 20 optimization iteration across all three attack setups.

In Fig. 1(b), we assess the visual fidelity limitations of images generated by SOTA GAN-based MIAs, specifically LOMMA (GMI), KEDMI, and PLG-MI, and compare their results to those produced by our SDM-based LOMMA (GMI) approach. For optimization, we perform 300 iterations for the SDM-based setup, 1,200 iterations for KEDMI and LOMMA (GMI), and 600 iterations for PLG-MI. We employ VGG16 and face.evoLve, both trained on the CelebA dataset, as the target and evaluation models, respectively. Additionally, we use the CelebA dataset as the public dataset to learn prior knowledge. For each attack setup, we select 1 of the 10 nearest neighbor images that exhibits the best visual fidelity as the final result and annotate the corresponding KNN distance relative to the private training data in the top right corner of the image.

## D ADDITIONAL EXPERIMENTAL RESULTS

### D.1 ADDITIONAL MAIN RESULTS

**Comparison with label-only classifier inversion.** In this setting, we use the SOTA label-only BREP-MI (Kahla et al., 2022) as the baseline for comparison. Kahla et al. (2022) introduce a boundary-repelling algorithm to search for representative samples by estimating the direction towards the target class's centroid using the predicted labels of the target model over a sphere. The quantitative results are presented in Tab. 5, with additional results in Tab. 9, and visualizations in Fig. 8. Across all evaluation metrics, our SDM-based BREP-MI consistently achieves better MI performance compared to the baseline, generating images with more representative features of the target class. Due to the more powerful capabilities of SDM, the generated images exhibit greater fidelity. These quantitative and visual results demonstrate that SDM-based BREP-MI effectively finds the optimal radius with fine granularity, outperforming the baseline method.

**Attacks Against SOTA model inversion defense methods.** We extended our evaluation to include state-of-the-art model inversion defense methods, such as BiDO-HSIC (Peng et al., 2022), NegLS (Struppek et al., 2024), and TL-DMI (Ho et al., 2024), comparing the baseline LOM (GMI)

Table 6: Model inversion performance against state-of-the-art defense methods. The target model M = VGG16 is trained on $\mathcal{D}_{pri}$ = CelebA, generative models are trained on $\mathcal{D}_{pub}$ = CelebA.

| Method | LOM (GMI) | | | | LOM (GMI) w/ SDM | | | |
|---|---|---|---|---|---|---|---|---|
| | Acc@1↑ | Acc@5↑ | KNN Dist↓ | FID↓ | Acc@1↑ | Acc@5↑ | KNN Dist↓ | FID↓ |
| No Def. | 63.48 | 86.07 | 1413.70 | 47.07 | 67.31 | 83.39 | 1382.15 | 44.86 |
| w/ TL-DMI | 37.77 | 64.54 | 1568.78 | 49.51 | 43.21 | 64.19 | 1555.60 | 26.52 |
| w/ NegLS | 24.88 | 51.73 | 1529.21 | 41.97 | 34.80 | 61.06 | 1517.75 | 183.41 |
| w/ BiDO-HSIC | 46.49 | 71.63 | 1535.47 | 46.24 | 40.08 | 60.65 | 1627.17 | 42.20 |

approach with our SDM-based LOM (GMI) method. The results, summarized in Tab. 6, show that the SDM-based approach outperforms the baseline across several metrics. Specifically, under TL-DMI, our method achieves better attack accuracy, KNN distance, and FID. Against NegLS, we observed improvements in attack accuracy and KNN distance, while for BiDO-HSIC, the SDM-based method demonstrated superior FID scores.

However, with NegLS, while attack accuracy improved, the FID score was lower compared to the baseline, and visual inspection revealed that the SDM-based approach generated distorted images with reduced visual quality. In contrast, under BiDO-HSIC, the SDM-based method produced more visually representative images, despite the attack accuracy and KNN distance being lower than the baseline, while still achieving an improved FID score. The inconsistency between visual quality and traditional metrics suggests that attack accuracy may not be a reliable indicator in some cases. Thus, more robust evaluation metrics are needed to better assess model inversion performance and reconcile discrepancies between visual and quantitative results.

### D.2 EXPERIMENTAL RESULTS WITH ADDITIONAL EVALUATION METRICS

We compare additional metrics—precision, recall, density, and coverage—as a supplement to the conventional metrics discussed in the main results section, providing further context. Although diversity is not the primary objective of MIAs, these metrics offer useful insights. The comparison with GMI and LOMMA (GMI) is shown in Tab 7, with PLG-MI in Tab 8, and BREP-MI in Tab 9.

### D.3 VISUALIZATION OF RECONSTRUCTED IMAGES

In this section, we present qualitative evidence to highlight the effectiveness of DDMI. For the white-box setting, Fig. 5 shows a visual comparison of reconstructed images from VGG16 and face.evoLVe models, both trained on the CelebA private dataset, with image priors learned from CelebA. Similarly, Fig. 6 provides a comparison of reconstructed images from VGG16 and face.evoLVe trained on the CelebA private dataset, but with image priors learned from the FFHQ dataset. Additionally, Fig. 7 displays a visual comparison of reconstructed images from the same models, with FFHQ as the source of image priors. For the black-box BREP-MI comparison, the visual results are provided in Fig. 8. In this case, the target model is VGG16 trained on the CelebA private dataset, while the image priors are learned from either the CelebA public dataset or the FFHQ dataset.

## E DISCUSSION

**Limitation.** One key limitation we found is the much higher dimensionality of the latent space in diffusion models compared to traditional GANs. In contrast to GANs, where the latent space is typically low-dimensional, the latent space in diffusion models matches the input space dimensions, creating challenges in extending DDMI to black-box attack methods like RLB-MI (Han et al., 2023) and BREP-MI (Kahla et al., 2022), which rely on black-box optimization.

For RLB-MI, the agent typically interacts with a DCGAN, which operates in a low-dimensional latent space (*e.g.,* 100 dimensions). This smaller latent space facilitates more efficient exploration and optimization within the *Markov decision process* (MDP). However, when DCGAN is replaced with a diffusion model, the latent space expands to much higher dimensions (*e.g.,* $64 \times 64 \times 3$), making it far more challenging for the agent to navigate and optimize within the MDP framework. Similarly, in BREP-MI, the challenge lies in identifying initial latent codes that generate images classified into

Table 7: Additional metrics used to evaluate inversion performance with GMI and LOMMA (GMI). The target model M = VGG16 or face.evoLVe, trained on $\mathcal{D}_{\text{pri}}$ = CelebA. Besides, generative models are trained on $\mathcal{D}_{\text{pub}}$ = CelebA or FFHQ.

| Target Model | Method | CelebA | | | | FFHQ | | | |
|---|---|---|---|---|---|---|---|---|---|
| | | Precision↑ | Recall↑ | Density↑ | Coverage↑ | Precision↑ | Recall↑ | Density↑ | Coverage↑ |
| **VGG16** | GMI | 0.0182 | 0.0041 | 0.1607 | 0.1244 | 0.0205 | 0.0051 | 0.1474 | 0.1065 |
| | w/ SDM | 0.0783 | 0.0568 | 0.6137 | 0.3027 | 0.0754 | 0.0575 | 0.5948 | 0.3003 |
| | LOMMA (GMI) | 0.0246 | 0.0082 | 0.2535 | 0.1516 | 0.0244 | 0.0032 | 0.2186 | 0.1346 |
| | w/ SDM | 0.0861 | 0.0575 | 1.0196 | 0.4833 | 0.0894 | 0.0695 | 1.0026 | 0.5069 |
| **face.evoLVe** | GMI | 0.0182 | 0.0041 | 0.1607 | 0.1244 | 0.0205 | 0.0051 | 0.1474 | 0.1065 |
| | w/ SDM | 0.0783 | 0.0568 | 0.6137 | 0.3027 | 0.0754 | 0.0575 | 0.5948 | 0.3003 |
| | LOMMA (GMI) | 0.0246 | 0.0082 | 0.2535 | 0.1516 | 0.0244 | 0.0032 | 0.2186 | 0.1346 |
| | w/ SDM | 0.0861 | 0.0575 | 1.0196 | 0.4833 | 0.0894 | 0.0695 | 1.0026 | 0.5069 |

Table 8: Additional metrics used to evaluate inversion performance with PLG-MI on different target models trained on $\mathcal{D}_{\text{pri}}$ = CelebA. Besides, conditional generative models are trained on $\mathcal{D}_{\text{pub}}$ = FFHQ.

| Method | VGG16 | | | | face.evoLVe | | | |
|---|---|---|---|---|---|---|---|---|
| | Precision↑ | Recall↑ | Density↑ | Coverage↑ | Precision↑ | Recall↑ | Density↑ | Coverage↑ |
| PLG | 0.0535 | 0.0010 | 0.8374 | 0.2733 | 0.0536 | 0.0453 | 0.7817 | 0.2960 |
| w/ SDM (ours) | 0.0631 | 0.1377 | 0.6630 | 0.3377 | 0.0608 | 0.1813 | 0.6141 | 0.3230 |

Table 9: Additional metrics used to evaluate inversion performance with BREP-MI. The target model M = VGG16 trained on $\mathcal{D}_{\text{pri}}$ = CelebA. Besides, generative models are trained on $\mathcal{D}_{\text{pub}}$ = CelebA or FFHQ.

| Method | CelebA | | | | FFHQ | | | |
|---|---|---|---|---|---|---|---|---|
| | Precision↑ | Recall↑ | Coverage↑ | Density↑ | Precision↑ | Recall↑ | Coverage↑ | Density↑ |
| BREP-MI | 0.0302 | 0.0140 | 0.0286 | 0.0435 | 0.0484 | 0.0496 | 0.0337 | 0.0563 |
| w/ SDM (ours) | 0.1216 | 0.0806 | 0.0868 | 0.1487 | 0.0728 | 0.0971 | 0.0365 | 0.0668 |

the target class. When a distribution gap exists between the public dataset used for prior knowledge and the private dataset being attacked, generating accurate initial latent codes becomes challenging.

**What are the potential future research directions?**

- **Extending DDMI to Black-Box Settings:** Addressing the limitation we identified, an important future research direction would be to extend DDMI to black-box settings. This would enable the method to work in more restricted environments where access to the internal parameters of models is limited.

- **Inversion-Specific Distillation:** A key challenge during diffusion distillation is effectively utilizing information from the target classifier. Exploring ways to more fully integrate this information into the distillation process remains an important area of research.

- **Generative CLIP Inversion:** As the first work to extend generative MIAs to CLIP models, several open questions remain. A key challenge is the evaluation, where collecting images of celebrities who are highly represented in the training data of open-source CLIP models, such as the LAION-400M dataset, is essential for accurately assessing the privacy leakage risks in CLIP. Moreover, future work should investigate how to more effectively utilize text data, model structure, and intermediate outputs during the inversion process.

- **Robust Privacy-Preserving Mechanisms for CLIP Models:** Our findings emphasize the pressing need for robust and effective privacy-preserving techniques during model training, especially for CLIP models, which remain largely underexplored in terms of privacy vulnerabilities. Addressing this gap would be crucial in developing safer models.

- **Improving Evaluation Metrics for Model Inversion Attacks:** We observed that the primary evaluation metric for MIAs, attack accuracy, does not always accurately reflect inversion success. Hence, developing more reliable metrics that capture the discriminative semantic information present in the samples is another important direction for future work.

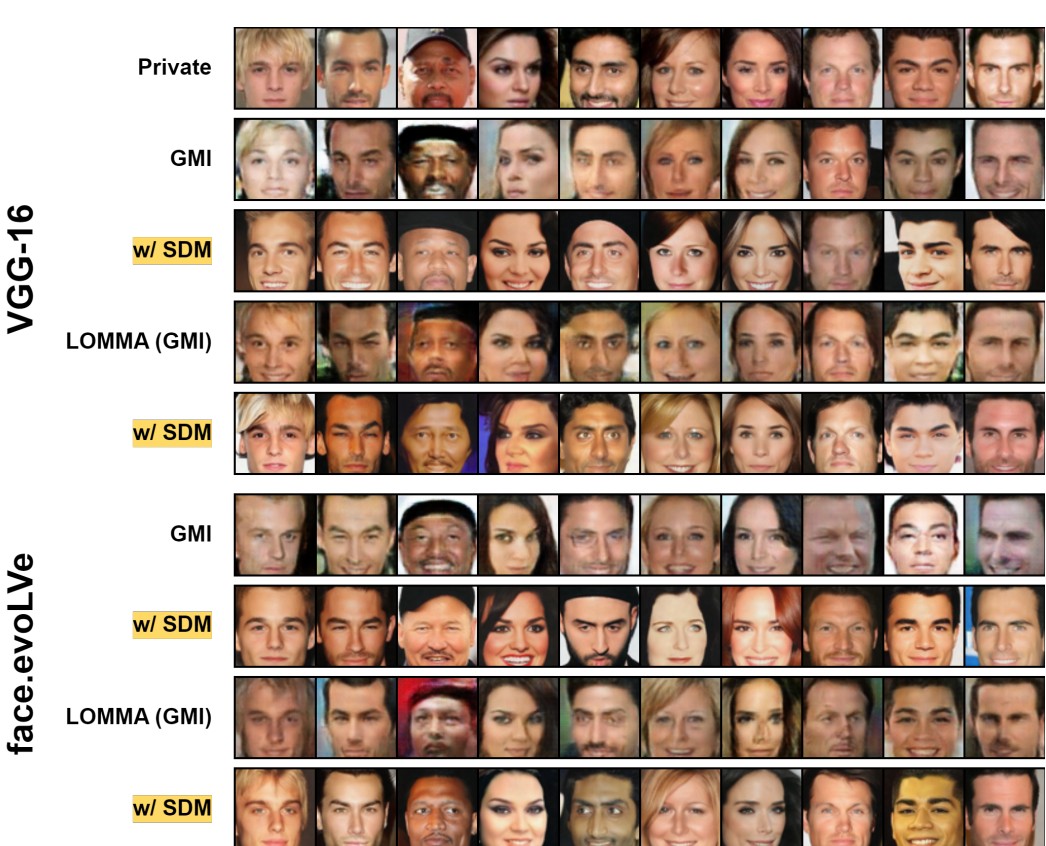

Figure 5: Visual comparison with GMI and LOMMA (GMI). We illustrate reconstructed samples for ten identities in $\mathcal{D}_{pri}$ = CelebA, generative models are trained on $\mathcal{D}_{pub}$ = CelebA. The target model M = VGG16 or face.evoLVe.

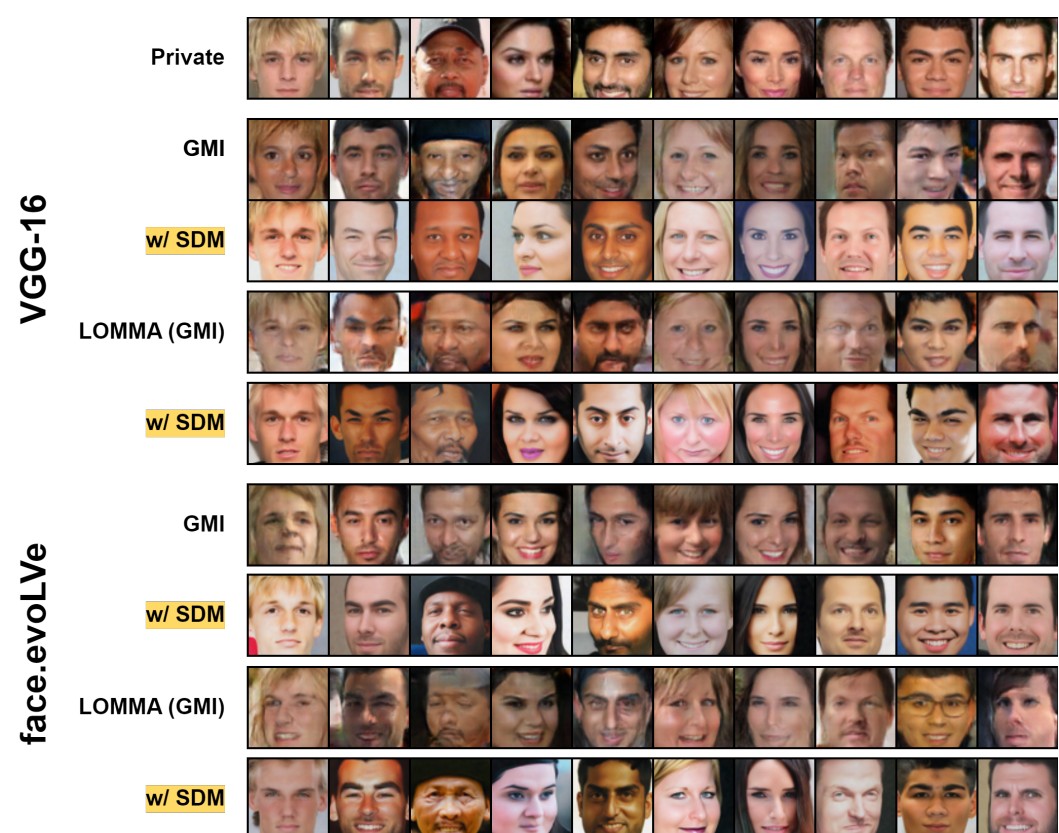

Figure 6: Visual comparison with GMI and LOMMA (GMI). We illustrate reconstructed samples for ten identities in $\mathcal{D}_{pri}$ = CelebA, generative models are trained on $\mathcal{D}_{pub}$ = FFHQ. The target model M = VGG16 or face.evoLVe.

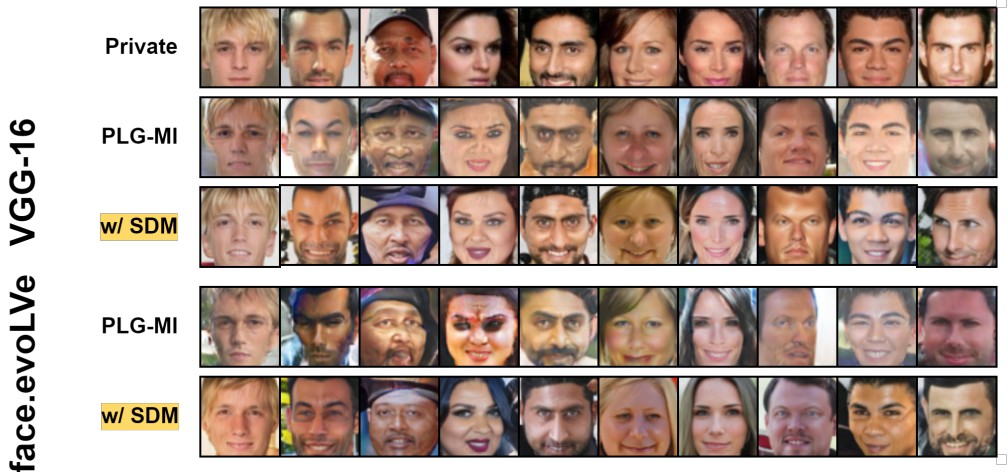

Figure 7: Visual comparison with PLG-MI. We illustrate reconstructed samples for ten identities in $\mathcal{D}_{pri}$ = CelebA, conditional generative models are trained on $\mathcal{D}_{pub}$ = FFHQ. The target model M = VGG16 or face.evoLVe.

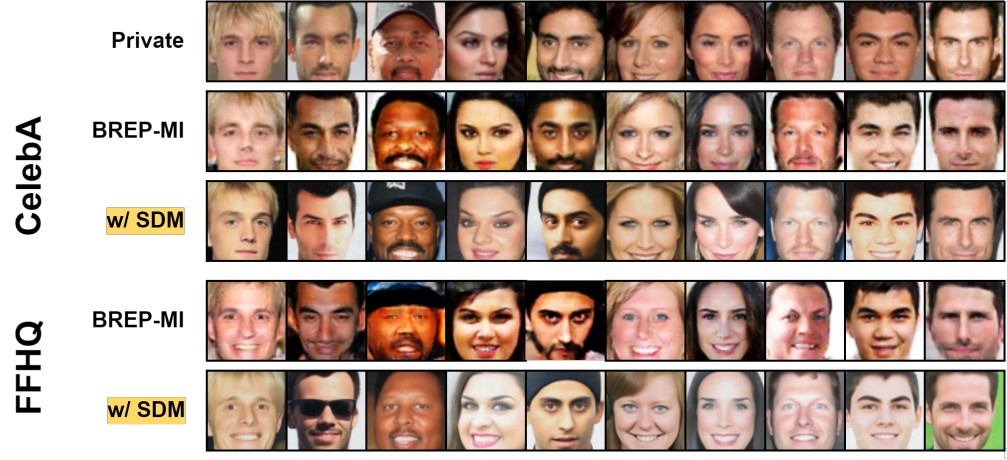

Figure 8: Visual comparison with BREP-MI. We illustrate reconstructed samples for ten identities in $\mathcal{D}_{\text{pri}}$ = CelebA, generative models are trained on $\mathcal{D}_{\text{pub}}$ = CelebA or FFHQ. The target model M = VGG16.

