# OpenReview forum: "Single-Step Diffusion Model-Based Generative Model Inversion Attacks"
_ICLR.cc/2025/Conference — Submitted to ICLR 2025_

### Official Review · Reviewer_kqH9 · 2024-11-03

**Soundness:** 2
**Presentation:** 3
**Contribution:** 3
**Rating:** 3
**Confidence:** 5

**Summary:**

This paper focuses on the advancement of generative model inversion attacks (MIAs). It substitutes the GAN prior with a single-step generator distilled from a pretrained multi-step diffusion model. Besides, this paper first explores CLIP model inversion. Experimental results demonstrate the enhanced performance compared to baselines.

**Strengths:**

+ **Good Writing**: The paper is well-structured and easy to follow.
+ **New Scenario**: The paper explores a new scenario of CLIP model inversion.

**Weaknesses:**

+ There are two “Sec.3.1” on the 243~244 lines.
+ The specific expression of $\mathcal{L}_{distill}$ in formula 8 is not explained in the paper.
+ There is a lack of high-resolution comparisons with the two attack methods, including Mirror [1] and PPA [2].

[1] An, Shengwei, et al. "Mirror: Model inversion for deep learning network with high fidelity." *Proceedings of the 29th Network and Distributed System Security Symposium*. 2022.

[2] Struppek, Lukas, et al. "Plug & play attacks: Towards robust and flexible model inversion attacks." *arXiv preprint arXiv:2201.12179* (2022).

**Questions:**

The trend in Fig 1(a) seems similar between lines. More explanation is expected.

---

> ### Author Response · Authors · 2024-11-23
> **Rebuttal by Authors**
>
> Thank you for your careful review of our paper and for pointing out these issues. We have provided detailed responses to your comments below.
>
> > W3. There are two “Sec.3.1”.
>
> Thank you for identifying the typographic errors. We have corrected the duplicate "Sec. 3.1" issue in the revised version of the paper. Your attention to detail is appreciated.
>
> > W2. The specific expression of $L_{distill}$ in formula 8 is not explained in the paper.
>
> We appreciate the reviewer’s helpful comments regarding the explanation of $L_{distill}$ in Eq. (8).
> The $L_{distill}$ term in Eq. (8) can represent any suitable distillation loss function. To address this concern and improve clarity, we will add explanations in the paper to clarify any ambiguity. Specifically, in Section 3.4, we instantiate $L_{distill}$ as $L_{SiD}$, which is tailored to our application.
>
> > Q1. The trend in Fig 1(a) seems similar between lines. More explanation is expected.
>
> In Fig 1(a), there is indeed a similar trend across all methods: attack accuracy generally increases as optimization iterations proceed. This trend reflects that samples are increasingly optimized to approximate private data with more iterations.
>
> However, our focus here is to reveal the instability issues present in the optimization process of the SOTA methods. While all methods show an upward trend, the SOTA methods display consistent fluctuations and higher variance in attack accuracy, indicating instability. In contrast, our SDM-based LOMMA (GMI) method demonstrates a steadier increase with minimal fluctuations, highlighting a more stable optimization process.

---

> > ### Comment · Reviewer_kqH9 · 2024-11-24
> >
> > My core question of **high-resolution comparisons** remains unresolved. Moreover, considering all the reviews from other reviewers, I think this article is a semi-finished work and not qualified to be accepted. Therefore, I have to lower my score.

---

### Official Review · Reviewer_MUun · 2024-11-03

**Soundness:** 2
**Presentation:** 2
**Contribution:** 2
**Rating:** 3
**Confidence:** 5

**Summary:**

The paper proposes a novel model inversion attack with single-step diffusion models. It highlight the instability and low visual fidelity of GANs in MIAs and propose to use diffusion models to overcome the issue. The attackers train a generator distilled by the diffusion model instead of directly training a GAN. The technique can strengthen existing attack methods such as GMI. Moreover, the paper firstly conducts a model inversion research on the multi-modal field, exploring the attacks in CLIP models.

**Strengths:**

+ The paper use a pre-trained diffusion model to distill a generator, which enhance the stability and the fidelity of the reconstructed images.
+ The paper is the first to explore privacy leakage on large-scale CLIP models leveraging MIAs.

**Weaknesses:**

+ The method of distilling a single-step generator has been extensively studied in prior work, which limits the novelty of this research. The DDMI approach primarily applies the diffusion distillation technique to model inversion attacks without introducing significantly new ideas.

+ The new investigation for CLIP model inversion is non-sense. First, the numerical results are so weak to prove the urgency of privacy security and the visual results is good mainly due to the inherent ability of the generator, lacking essential experiments for more convincing evidence. Nonetheless, the proposed method does not perform better than StyleGAN-based generative CLIP inversion, which means no superiority. According to the explanation for low performance, more improvement and evaluation should be made to support the research of CLIP model inversion.

+ The experimental results are not sufficient. In Table 1, the classical methods PPA [1], KEDMI [2], PLGMI [3] and LOMMA(KEDMI) [4] are not compared. For target models, the series of ResNet [5] models are usually evaluated in the previous methods, which should be included in the experiments. In addition, there is a lack of comparison with StyleGAN in experimental settings other than CLIP.

+ The experiment setup is too weak. For example, dividing CelebA into public and private parts is a quite easy scenario with low distributional shift, where PLGMI can achieve nearly 99% accuracy. Investigation on such scenario is non-sense to prove the superiority of your method, much less the insignificant improvement compared with LOMMA. Moreover, the division of CelebA is not introduced in this paper.

+ The experiments of defenses in Table 6 shows that the attack results are worser than the case without SDM with the BiDO defense.

[1] Struppek L, Hintersdorf D, Correia A D A, et al. Plug & play attacks: Towards robust and flexible model inversion attacks[J]. arXiv preprint arXiv:2201.12179, 2022.

[2] Si Chen, Mostafa Kahla, Ruoxi Jia, and Guo-Jun Qi. Knowledge-enriched distributional model inversion attacks. In ICCV, 2021.

[3] Yuan X, Chen K, Zhang J, et al. Pseudo label-guided model inversion attack via conditional generative adversarial network[C]//Proceedings of the AAAI Conference on Artificial Intelligence. 2023, 37(3): 3349-3357.

[4] Nguyen N B, Chandrasegaran K, Abdollahzadeh M, et al. Re-thinking model inversion attacks against deep neural networks[C]//Proceedings of the IEEE/CVF Conference on Computer Vision and Pattern Recognition. 2023: 16384-16393.

[5] He, K., Zhang, X., Ren, S., Sun, J.: Deep residual learning for image recognition. In: Proceedings of the IEEE Conference on Computer Vision and Pattern Recognition, pp. 770–778 (2016)

**Questions:**

According to the ablation study, the prior loss specified in your method brings negative improvement. Why do you keep this innovation point in your method?

---

> ### Author Response · Authors · 2024-11-24
> **Rebuttal by Authors**
>
> Thank you for your time in reviewing our work and for your comments. Please see our responses to your comments and questions below.
>
> > W1. The method of distilling a single-step generator has been extensively studied in prior work, which limits the novelty of this research.
>
> While the technique of distilling a single-step generator has indeed been studied in prior work, our contribution lies in its novel application within the context of model inversion attacks (MIA), which has not been explored before. Specifically, we introduce inversion-specific distillation process in Section 3.3, and embed it within a new framework diffusion distillation MIAs (DDMI) that leverages diffusion models to significantly enhance generative MIAs.
>
> This combination of techniques and their application to MIA represents a meaningful innovation, as it not only demonstrates how single-step generators can improve attack effectiveness but also establishes a new perspective on evaluating privacy vulnerabilities through advanced generative models.
>
> > W2. The new investigation for CLIP model inversion is non-sense.
>
> We hypothesize that utilizing more detailed templates can improve numerical performance. Our additional experiments show that detailed templates reduce the overall KNN distance, as illustrated in the table below, where we use ViT-L-14 as the vision model in three methods.
>
> | Method         | Simple template | Detailed template |
> |----------------|-----------------|-------------------|
> | CLIPInversion  | 1.1018          | 0.9747            |
> | w/ SDM         | 0.9099          | 0.8989            |
> | w/ StyleGAN    | 0.8829          | 0.8730            |
>
> Furthermore, we suggest that the numerical results are influenced by the frequency of target entities in the LAION-400M dataset. The vision model was trained on the LAION-400M dataset, which contains a diverse range of entities, and most target entities in the FaceScrub dataset has low frequency.
>
> > W3. The experimental results are not sufficient.
>
> As your comments are partially similar to those of Reviewer mTRw, please refer to our response to Reviewer mTRw‘s W1 for further details.
>
> > W4. The experiment setup is too weak.
>
> - Regarding the statement that “investigation on such a scenario is non-sense to prove the superiority of your method,” we would like to clarify that using CelebA as the public dataset is a commonly adopted benchmark setup in previous works, such as GMI [r1], KEDMI [r2], PPA [r3] and LOMMA [r4], even though PLG achieves perfect attack accuracy. Our objective was to demonstrate the improvements achieved by integrating SDM, which we did by comparing performance before and after this integration.
>
> - Besides, FFHQ as public dataset is also included to address scenarios with higher distributional shifts, allowing for a more robust comparison, particularly with PLGMI. Our paper already includes experiments using FFHQ as the public dataset, and the results with FFHQ are presented in Table 3. This setup introduces greater distributional shift and provides a more challenging scenario for evaluating our method.
>
> - Regarding the division of CelebA, we follow default setup in the standard MIA literature, which divides CelebA into two non-overlapping sets. We will explicitly include the dataset division in our experiment setup. Thanks for pointing this issue.
>
>
> > W5. The experiments of defenses in Table 6 shows that the attack results are worser than the case without SDM with the BiDO defense.
>
> It is indeed true and acceptable that every method can exhibit failure cases, especially when faced with robust defenses like BiDO. The SDM-based method features a more powerful and higher-quality generator, resulting in significantly better image quality compared to traditional methods (refer to Fig.5 in our revised paper). The generated images are more representative and reveal more privacy features. From the perspective of visual results, SDM can still more clearly demonstrate the overall characteristics of the data distribution, thereby exposing more potential privacy information.
>
> However, from a data perspective, we believe that current metrics may not effectively measure MI performance. While SDM shows lower performance than the baseline in attack accuracy and feature distance, it exhibits an opposite trend in the FID metric.
>
> One possible explanation for this is that SDM's generative objective may place greater emphasis on the consistency of the overall data distribution rather than precisely reproducing the fine-grained details of specific inputs. As a result, the samples generated by SDM tend to reflect the average characteristics of the distribution. Although they achieve higher visual quality, they may be limited in capturing specific target privacy details and improving attack accuracy.

---

> > ### Comment · Reviewer_MUun · 2024-11-24
> >
> > In response to W2, the effect of using SDM is still not as good as StyleGAN, so why use SDM?
> > In response to W3, the authors refuse to perform additional experiments, such as PPA, focusing on high-resolution settings.
> > Therefore, I still maintain my score.

---

> ### Author Response · Authors · 2024-11-24
> **Remaining Responses to Reviewer MUun**
>
> > Q1. According to the ablation study, the prior loss specified in your method brings negative improvement. Why do you keep this innovation point in your method?
>
> The prior loss was proposed as part of the diffusion distillation framework to explore its potential effects on performance. However, it is not a mandatory component for optimization in our method. As shown in our ablation study, the prior loss brings negative improvement in some cases, which is why we excluded it from the main optimization.  It is similar to how discriminator loss is not utilized in PPA [r3], discussed in its ablation study. Its inclusion was intended to examine its potential influence and provide a more comprehensive understanding of its role, rather than to position it as an essential element of our approach.
>
> ---
> **References**:
>
> [r1] Zhang et al. "The secret revealer: Generative model-inversion attacks against deep neural networks." In CVPR, 2020.
>
> [r2] Chen et al. "Knowledge-enriched distributional model inversion attacks." In ICCV, 2021.
>
> [r3] Struppek et al. "Plug & play attacks: Towards robust and flexible model inversion attacks." In ICML, 2022.
>
> [r4] Nguyen et al. "Re-thinking model inversion attacks against deep neural networks." In CVPR, 2023.

---

### Official Review · Reviewer_tF8d · 2024-11-03

**Soundness:** 3
**Presentation:** 3
**Contribution:** 2
**Rating:** 6
**Confidence:** 3

**Summary:**

The study explores generative model inversion attacks (MIAs) using diffusion models to address the limitations of GAN-based inversion attacks. These limitations include instability during the optimization process and low fidelity in reconstructed samples. By incorporating a single-step generator distilled from pretrained diffusion models, the paper presents a novel framework termed Diffusion Distillation MIAs (DDMI). This approach is applied not only to traditional classification models but also to the multimodal CLIP models, uncovering significant privacy risks.

**Strengths:**

1. Extending MIAs to explore privacy leakage in CLIP models marks a significant original contribution, venturing into an area that has not been extensively studied before. This application highlights the inherent privacy risks in large-scale multimodal models and sets a precedent for future research in this domain.

2.The paper successfully demonstrates that DDMI outperforms traditional GAN-based methods in achieving higher fidelity and stability during the inversion process.

**Weaknesses:**

1. In my opinion, "Distill the Multi-Step Diffusion Model into a Single-Step Generator" has no much innovation, similar to other One-step Diffusion papers.

2. The paper could benefit from including a broader array of evaluation metrics that could capture other aspects of model performance and attack effectiveness, such as computational efficiency, robustness to adversarial defenses, or qualitative user studies.

**Questions:**

1. In the experiments, why is there no comparison of computational overhead between gan-based and diffusion-based methods?

2. Why there is no analysis of the robustness of adversarial defenses in the experiment or varience experiment?

---

> ### Author Response · Authors · 2024-11-25
> **Rebuttal by Authors**
>
> Sincerely thank you for your constructive comments and generous supports! Please see our responses to your comments and suggestions below.
>
> > W1. "Distill the Multi-Step Diffusion Model into a Single-Step Generator" has no much innovation, similar to other One-step Diffusion papers.
>
> As your comments are similar to those of Reviewer MUun, please refer to our response to Reviewer MUun‘s W1 for further details.
>
> > W2. The paper could benefit from including a broader array of evaluation metrics that could capture other aspects of model performance and attack effectiveness, such as computational efficiency (Q1), robustness to adversarial defenses, or qualitative user studies.
>
> We would also like to clarify that experiments on robustness to adversarial defense are included in our paper. Specifically, we discuss the results in Appendix D.1, with qualitative results presented in Table 6. As for computational efficiency and qualitative user studies, we will incorporate these metrics into the revised version of our manuscript.
>
> > Q2. Why there is no analysis of the robustness of adversarial defenses in the experiment or variance experiment?
>
> For experiments and analysis of the robustness of adversarial defenses, please refer to the response to W2.
>
> For variance experiment, we have conducted additional experiments to include standard deviation of main metrics and please refer to the general reply.

---

### Official Review · Reviewer_mTRw · 2024-11-04

**Soundness:** 2
**Presentation:** 3
**Contribution:** 2
**Rating:** 5
**Confidence:** 4

**Summary:**

This work studies techniques for improving generative Model Inversion Attacks (MIA) in white-box settings. The major contributions of this work are:

1) In the context of model inversion attacks, this paper identifies key limitations of GAN-based generative MIAs, namely unstable optimization and low-fidelity sample reconstruction (Fig. 1). To address these challenges, the authors propose a framework called DDMI (Diffusion Distillation Model Inversion Attacks).

2) The proposed DDMI framework obtains notable improvements compared to GMI and LOMMA (GMI) (Table 1), and PLG-MI (Table 3).

3) This work also explores MIAs for CLIP models.

**Strengths:**

1) This paper is written well and it is easy to follow.

2) The proposed method obtains acceptable improvements over prior methods.

**Weaknesses:**

1) **Poor selection of baseline methods for comparison in Section 4.2.** GMI is a relatively older work (although it lays foundation for modern generative MIAs) and generally has poor MI accuracy.

- Authors should consider using more advanced baselines such as KEDMI/ LOMMA+KEDMI / PLGMI for SOTA comparison in Table 1.

2) **No empirical evidence/ theoretical grounds for strong claims made in Section 3.1.** Specifically, it is unclear how the authors arrive at the statement, “We attribute these limitations to inherent flaws in GANs” in the context of MIAs.

- Such claims are strong and need clear theoretical or empirical support. Additionally, this statement conflicts with the results reported in the SDM-based vs. StyleGAN-based generative CLIP inversion (Page 10).

- The claim about unstable optimization under GAN-based MIAs requires careful study. The qualitative results shown in Fig. 1(a) alone are insufficient. It would be helpful to explore whether this instability is due to the generative model's architecture, the training objective, the data, or the model quality. One approach to examine this further is to use a checkpoint of the one-step diffusion model with a comparable FID to the GAN, then reproduce Fig. 1(a). This could provide more clarity on the claim.

3) **User studies are necessary to show the quality of privacy reconstruction.** Since this work focuses on private data reconstruction, especially highlighting fidelity, it is important to conduct user study to understand the improvements (See [A, B]).

4) Error bars/ Standard deviation for experiments are missing.

5) In Supplementary D, authors should consider including LOKT [A], as it is the state-of-the-art label-only MIA and significantly outperforms BREP-MI.


Minor:

Table 1 VGG16 w/SDM / CelebA top5 accuracy arrow should be $\uparrow$.

=============

[A] Nguyen, Bao-Ngoc, et al. "Label-only model inversion attacks via knowledge transfer." Advances in Neural Information Processing Systems 36 (2024).


[B] [MIRROR] An, Shengwei et al. MIRROR: Model Inversion for Deep Learning Network with High Fidelity. Proceedings of the 29th Network and Distributed System Security Symposium.

**Questions:**

Overall I enjoyed reading this paper. But in my opinion, the weaknesses of this paper outweigh the strengths. But I’m willing to change my opinion based on the rebuttal.


Please see Weaknesses section above for a list of all questions.


======

Post-rebuttal

Thank you authors for the extensive rebuttal and performing additional experiments. I have read all the reviews and the rebuttal. The rebuttal addresses some of my concerns (PLGMI, Error bars for experiments, clarification reg. the use of inversion-specific GANs in the proposed framework). I acknowledge the notable performance gains achieved using your proposed framework and it can be valuable to the community. However, the core questions still remain as acknowledged by the authors in the rebuttal (No empirical evidence/ theoretical grounds for strong claims in Sec 3.1, user studies).

Therefore, I’m afraid I’m unable to increase my rating, as I feel the paper is not ready for publication in its current form. I have suggested some directions for improving Section 3.1 (See my review). Another way to think more deeply about this problem is as follows: **Model inversion attacks aim to search for high-likelihood samples under a target identity. Why, then, does your proposed framework yield significantly better results in this search compared to existing methods? Investigating these questions could provide a more rigorous analysis and a deeper understanding of your framework.**

Overall, this work has good potential, and I wish you the best as you continue to refine it. I also encourage you to address high-resolution MIAs, as suggested by other reviewers.

Thanks again for the extensive rebuttal.

---

> ### Author Response · Authors · 2024-11-24
> **Rebuttal by Authors**
>
> Thank you for your constructive comments! Please see our detailed responses to your comments and suggestions below.
>
> > W1. Poor selection of baseline methods for comparison in Section 4.2.
>
> - First, we acknowledge that KEDMI and LOMMA (KEDMI) represent strong baselines. However, KEDMI utilizes inversion-specific GAN models to derive the generator, which contrasts with our goal of using a plug-and-play generator that is applicable across various MIA frameworks. This distinction is clearly demonstrated in our experiments, where we evaluate performance using both the original generators and our single-step (conditional) diffusion model across multiple methods. By comparison, KEDMI relies on an inversion-specific GAN architecture, making it challenging to evaluate under the same plug-and-play framework.
>
> - That said, if an absolute performance comparison is desired, comparing conditional diffusion model derived from inversion-specific distillation with KEDMI might be more appropriate. Would you consider this comparison suitable?
>
> - We would also like to clarify that PLGMI is included as a baseline in our paper. Specifically, we discuss PLGMI in Section 4.2.1, with qualitative results presented in Table 3 and additional metrics in Table 8.
>
> > W2. No empirical evidence/ theoretical grounds for strong claims made in Section 3.1.
>
> - We agree that the statement, “We attribute these limitations to inherent flaws in GANs,” requires a more precise and well-supported explanation. Our motivation for introducing diffusion models is to leverage the rapid advancements and proven strengths of these models in generating high-fidelity images. Inspired by these developments, we aim to explore their potential as more robust tools for investigating model inversion attacks (MIAs). We will revise the paper to provide a clearer and more solid motivation for this transition.
>
> - Regarding the claim about instability in GAN-based MIAs, we recognize that the evidence provided in Fig. 1(a) alone may not be sufficient to justify such a strong conclusion. Based on experimental observations and further analysis, we hypothesize that the observed instability arises from the limited quality and capabilities of the GANs employed in these methods. We will revise our discussion to present an evidence-backed analysis of this aspect.
>
> - For the suggestion to include LOKT as a baseline, we acknowledge its significance as a state-of-the-art label-only MIA. However, similar to KEDMI, LOKT depends on a GAN-specific learning framework (T-ACGAN), making it less suitable for a direct comparison with our approach. Nevertheless, we appreciate the suggestion and will clarify this distinction in our revised manuscript.
>
> - For the standard deviation of our main experiments, please see our general reply.

---

> ### Comment · Reviewer_mTRw · 2024-11-26
> **[Reviewer mTRw] Response**
>
> Thank you authors for the extensive rebuttal and performing additional experiments. I have read all the reviews and the rebuttal. The rebuttal addresses some of my concerns (PLGMI, Error bars for experiments, clarification reg. the use of inversion-specific GANs in the proposed framework). I acknowledge the notable performance gains achieved using your proposed framework and it can be valuable to the community. However, the core questions still remain as acknowledged by the authors in the rebuttal (No empirical evidence/ theoretical grounds for strong claims in Sec 3.1, user studies).
>
> Therefore, I’m afraid I’m unable to increase my rating, as I feel the paper is not ready for publication in its current form. I have suggested some directions for improving Section 3.1 (See my review). Another way to think more deeply about this problem is as follows: **Model inversion attacks aim to search for high-likelihood samples under a target identity. Why, then, does your proposed framework yield significantly better results in this search compared to existing methods? Investigating these questions could provide a more rigorous analysis and a deeper understanding of your framework.**
>
> Overall, this work has good potential, and I wish you the best as you continue to refine it. I also encourage you to address high-resolution MIAs, as suggested by other reviewers.
>
> Thanks again for the extensive rebuttal.

---

> > ### Author Response · Authors · 2024-11-27
> >
> > Thank you for taking the time to review our rebuttal and for recognizing the potential of our work. We sincerely appreciate your constructive and thoughtful feedback.

---

### Official Review · Reviewer_KSUH · 2024-11-04

**Soundness:** 3
**Presentation:** 3
**Contribution:** 3
**Rating:** 5
**Confidence:** 4

**Summary:**

This paper introduces DDMI (diffusion distillation model inversion attacks), a novel approach that improves upon traditional GAN-based model inversion attacks by using distilled single-step diffusion models. The method shows superior performance in reconstructing private training data from both classification and CLIP models, demonstrating better visual quality, stability, and attack success rates compared to existing methods. Beyond technical improvements, the work is significant as the first to explore privacy vulnerabilities in CLIP models, highlighting urgent needs for better privacy protection in machine learning systems.

**Strengths:**

1. First attempt to explore CLIP model vulnerabilities
2. Good visualization of results

**Weaknesses:**

1. The core idea of replacing GANs with diffusion models have limited technical depth without fundamental insights into privacy mechanisms.
2. No theoretical gurantee of privacy such as Differential privacy or other theoretical foundations.

**Questions:**

1. How does this method fundamentally advance our understanding of privacy vulnerabilities beyond showing that better generative models lead to better attacks?
2. Why choose SiD specifically for distillation? What are the trade-offs compared to other distillation methods?

---

> ### Author Response · Authors · 2024-11-23
> **Rebuttal by Authors**
>
> Thank you for taking the time to review our work. We have provided detailed responses to your comments below.
>
> > W1. The core idea of replacing GANs with diffusion models have limited technical depth without fundamental insights into privacy mechanisms.
>
> According to the current theoretical understanding of model inversion attacks (MIAs) [r1], the presence of a powerful "common generator" is critical for the success of such attacks. Our proposed method, DDMI, introduces a novel perspective by leveraging diffusion models as an advanced tool in MIA research. Unlike GAN-based approaches, diffusion models, as likelihood-based methods, provide superior coverage of the data distribution. This capability enables diffusion models to learn a more effective "common generator," enhancing the MIA framework’s ability to reconstruct sensitive information with greater detail and accuracy. By adopting this approach, our work contributes to advancing the technical foundation of MIAs.
>
> > W2. No theoretical gurantee of privacy such as Differential privacy or other theoretical foundations.
>
> Theoretical guarantees of privacy, such as differential privacy, are typically associated with defense strategies designed to mitigate attack risks. Since our work aims to explore vulnerabilities and understand the capabilities of model inversion attacks, establishing theoretical privacy guarantees is outside the scope of this research. We appreciate your feedback and hope this clarification provides a clearer perspective on our study’s objectives.
>
> > Q1. How does this method fundamentally advance our understanding of privacy vulnerabilities beyond showing that better generative models lead to better attacks?
>
> Currently, generative MIAs serve as an empirical method for assessing privacy vulnerabilities in machine learning models. While previous MIA methods have shown strong performance in terms of specific metrics, the visual fidelity of recovered samples has generally remained limited. This limitation affects the practical effectiveness of these methods in accurately assessing privacy risks.
>
> Our approach advances the field by introducing a powerful single-step diffusion model, enhancing the capacity of generative MIAs to produce high-fidelity reconstructions. This improvement provides a more robust evaluation of privacy vulnerabilities in machine learning models, enabling a deeper understanding of the potential risks that such models may pose. Our work goes beyond simply demonstrating that improved generative models yield better attacks; it provides a concrete method for more effectively probing and evaluating model privacy in a meaningful way.
>
> > Q2. Why choose SiD specifically for distillation? What are the trade-offs compared to other distillation methods?
>
> Our goal is to enhance the effectiveness of model inversion attacks by leveraging a powerful diffusion model without introducing significant computational overhead. We chose SiD for distillation because it achieves state-of-the-art performance among diffusion distillation methods (up to the time of submission). Notably, SiD enables the student model to surpass the teacher model in generation quality, making it an exceptionally suitable choice for our approach.
>
> ---
> **References**:
>
> [r1] Wang et al. "Variational Model Inversion Attacks." In NeurIPS, 2021.

---

> > ### Comment · Reviewer_KSUH · 2024-12-02
> >
> > Thank you for the feedback. After careful consideration of the rebuttal and all reviewer comments, I maintain my original rating for the following reasons:
> >
> > 1. While the rebuttal demonstrated performance improvements, the work still lacks fundamental insights into privacy mechanisms beyond showing that better generative models yield better attacks. The core questions about how this method advances our theoretical understanding of privacy vulnerabilities remain inadequately addressed.
> > 2. The absence of theoretical privacy guarantees, though acknowledged as out of scope, represents a significant limitation for a paper focused on privacy vulnerabilities.
> > 3. Missing high-resolution comparisons as suggested by Reviewer kqH9
> >
> > These fundamental limitations suggest the work would benefit from further development before meeting ICLR's acceptance criteria.

---

### Author Response · Authors · 2024-11-25
**General Reply by Authors**

**General response:**
We thank all reviewers for their thoughtful suggestions on our submission. We address a common point in this response.

> Re: The variance experiments/standard deviation.

We have conducted additional main experiments to derive standard deviation and show the superiority of our method. For the experimental setup, we use the same attack parameters stated in our paper. We conduct three additional experiments by choosing different seeds for each attack setting to derive standard deviation.

We report top-1 attack accuracy (Acc@1) with standard deviation, top-5 attack accuracy (Acc@5) and KNN distance (KNN Dist) as detailed below:


|                  |               | CelebA           |                   |                       | FFHQ             |                 |                       |
|------------------|---------------|------------------|-------------------|-----------------------|------------------|-----------------|-----------------------|
| **Target Model** | Method        | Acc@1$\uparrow$  | Acc@5$\uparrow$   | KNN Dist $\downarrow$ | Acc@1$\uparrow$  | Acc@5$\uparrow$ | KNN Dist $\downarrow$ |
| **VGG16**        | LOMMA (GMI)   | 73.72            | 92.21             | 1316.63               | 54.39            | 79.44           | 1437.09               |
|                  | w/ SDM (ours) | 83.48 $\pm$ 0.14 | 94.25             | 1224.95               | 65.76 $\pm$ 0.44 | 87.02           | 1382.39               |
| **face.evoLVe**  | LOMMA (GMI)   | 80.21            | 94.79             | 1270.79               | 62.77            | 85.15           | 1406.75               |
|                  | w/ SDM (ours) | 87.21 $\pm$ 0.17 | 95.49             | 1173.37               | 72.11 $\pm$ 0.34 | 90.66           | 1343.41               |

---

- **We have made every effort to clarify and address the concerns that we could reasonably resolve. However, we understand that some issues may remain unresolved, and we do not expect this submission to necessarily meet the acceptance criteria at this stage. We thank all reviewers for their constructive suggestions and thoughtful comments, and we welcome any additional feedback that could further help us refine our work. We will continue to improve our manuscript based on the valuable insights provided.**

---

### Meta-Review · Area_Chair_yzHx · 2024-12-12

**Metareview:**

This paper explores multimodal privacy attacks on CLIP models using a distilled diffusion model.

The proposed method can work well, but there is a lack of explanation as to why, which is crucial for improving the community’s understanding of privacy in ML. Furthermore, the use of a distilled diffusion model as opposed to a GAN is not particularly novel, as distilled diffusion models are quite a standard technique these days. Merely applying them to MIAs is not a sufficiently novel contribution for ICLR.

However, the authors should not overly focus on this aspect when improving their work, as more pressing issues included an experimental evaluation that was missing highly relevant baselines and general depth. As such, I am recommending rejection.

**Additional Comments On Reviewer Discussion:**

The main points of improvement raised by reviewers were: the method of distilling a diffusion model for model inversion seems straightforward and lacking innovation; the experiments were lacking some relevant baselines; the experimental evaluation was generally lacking depth; some claims in Sec 3.1 were unsupported.

The authors made progress on including relevant baselines brought up by reviewers and tried to better support the claims in Sec 3.1. However, the depth of experimental analysis was still fairly shallow. The authors acknowledged that some additional experiments and justification are needed to back up their claims and will look to improve that aspect in a future submission.

---

### Decision · Program_Chairs · 2025-01-22

Reject